# Unsupervised Classification of Absorbing Aerosols Detected by the Single Particle Soot Photometer

Aaryan Doshi[1] and Kara D. Lamb[2]

[1]Stanford University, Palo Alto, CA, USA, 94305
[2]Department of Earth and Environmental Engineering, Columbia University, New York, NY, USA, 10027

**Correspondence:** Aaryan Doshi (adoshi25@stanford.edu) and Kara D. Lamb (kl3231@columbia.edu)

**Abstract.** The Single Particle Soot Photometer (SP2) detects refractory aerosol particle mass on a single-particle basis via laser-induced incandescence (L-II). While the SP2 has traditionally been used to quantify black carbon aerosol mass in the atmosphere, the instrument is increasingly being used to detect and quantify other types of absorbing aerosols, such as mineral dust or anthropogenically-sourced iron oxide aerosols. Quantifying the mass loadings and emission sources of absorbing

aerosols in the atmosphere is important for understanding their role in the climate cycle. Supervised machine learning algorithms have shown potential to classify different types of aerosols from L-II signals, but these methods are sensitive to instrument configuration and require training datasets generated from laboratory samples, which do not generalize well to ambient atmospheric aerosols. Here we explore the effectiveness of unsupervised machine learning, including principal component analysis (PCA) and a variational autoencoder (VAE), applied directly to L-II signals from the SP2 in order to classify

different types of absorbing aerosols. The VAE is a deep learning architecture that compresses L-II signals into a bottleneck latent representation and reconstructs an output as similar as possible to the input signal, thereby reducing dimensionality. We apply PCA and the VAE to a dataset comprised of laboratory samples of materials that show detectable incandescence in the SP2, including fullerene soot (as a proxy for black carbon), coated fullerene soot, coal fly ash, mineral dust, volcanic ash, hematite, and magnetite. We explore optimal latent representations of L-II signals to maximize separability of different

aerosol classes by varying the size of the latent representation, and find that a latent representation of 3 allows us to capture the majority of the information in the L-II signals relevant for identifying different types of absorbing aerosols. We demonstrate that unsupervised machine learning is a promising method for identifying distinct populations of aerosols detected by the SP2 and for quantifying the similarity of different types of aerosols in terms of their response in the SP2.

## 1 Introduction

Atmospheric aerosols originate from a multitude of sources: primarily from natural sources such as wind-blown dust from deserts, sea salt from the oceans, and smoke emitted from forest fires, and from anthropogenic sources such as the combustion of fossil fuels. Detection of these aerosols is essential to understanding the impact they have on the climate. While some aerosols such as sulfate aerosols from volcanoes can have a cooling effect by blocking out sunlight, other types of aerosols absorb light from the sun, heating up the atmosphere locally and substantially contributing to the overall warming of climate

(Bond et al., 2013; Baumgardner et al., 2012). Absorbing aerosols such as black carbon, brown carbon, and mineral dust are important short-lived climate forcers, with significant direct climate radiative effects. Quantifying the atmospheric abundance of these aerosols requires *in situ* observations in order to determine their emissions, sources, and lifetime in the atmosphere.

The Single Particle Soot Photometer (SP2), shown in Figure 1, is the state-of-the-art instrument for detecting refractory black carbon (rBC) in the atmosphere (Stephens et al., 2003). For the past two decades, the SP2 has been used in numerous

ground-based and airborne field studies to measure the atmospheric abundance of rBC (e.g. Schwarz et al., 2006; Moteki et al., 2014; Lamb et al., 2018), an aerosol sourced from incomplete combustion that has important implications for climate (Bond et al., 2013). Increasingly, the SP2 is being used to detect a more diverse spectrum of light-absorbing refractory aerosols, such as mineral dust and anthropogenically-sourced iron-oxide aerosols (Moteki et al., 2017; Liu et al., 2018; Lamb, 2019).

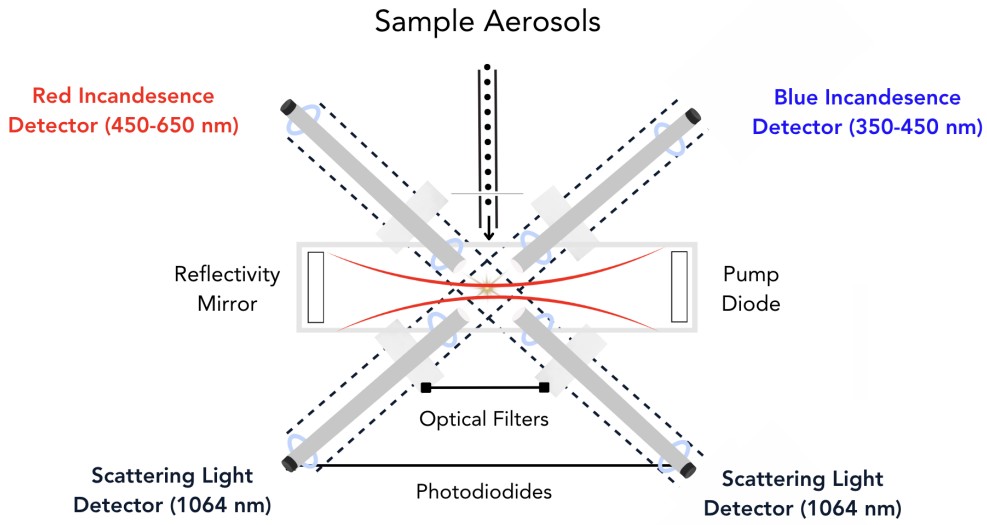

**Figure 1. A schematic of the SP2 instrument.** Aerosols sampled by the SP2 are introduced into the cavity of an ND:YAG laser (the beam is indicated as red curved lines), where they scatter light and potentially incandesce in the laser beam. Four photomultiplier tubes (indicated in the figure as the red detector, blue detector, and scattering light detectors) acquire time series of the light scattered or emitted by the aerosol particles as they interact with the laser beam. Figure adapted from Schwarz et al. (2006).

The SP2 uses laser-induced incandescence (L-II) to quantify refractory aerosol mass on a single particle basis (Stephens

et al., 2003). The SP2 samples aerosols in the sub-micron range by pulling particles into the cavity of an ND-YAG laser (1064 nm). As particles pass through the laser beam, if they have a sufficient absorption cross-section at 1064 nm, they will heat up and incandesce. Here we use observations from the NOAA SP2 instrument (Schwarz et al., 2006, 2010). In its typical configuration, the NOAA SP2 acquires signals on 4 detectors with a 5 MHz acquisition rate as the aerosols pass through the center of the laser beam. Two channels detect light scattered by particles as they pass through the center of the ND-YAG laser

using an avalanche photo-diode, and two channels detect light emitted by particles during incandescence using photomultiplier tubes (PMT) that measure visible light in two spectral bands: a narrow band PMT with peak sensitivity at 420 nm (350-450

nm) that we refer to as the "blue" incandescent channel, and a broadband PMT with peak sensitivity at 630 nm (450-650 nm) that we refer to as the "red" incandescent channel. One of the scattering channels has a position-sensitive detector and is used to determine the position of the particle relative to the center of the laser beam, as has been described in detail in Gao et al. (2007).

The position sensitive detector is used to determine the center point of the laser in order to derive information about the coating state of rBC particles by using the leading-edge-only fitting method and assuming Mie core-shell theory (Gao et al., 2007). The L-II signal associated with each aerosol detected by the SP2 therefore consists of time series from these 4 detection channels, which provide information about how the particle scatters and emits light as it passes through and (potentially) evaporates in the laser beam.

The SP2 is typically used in both ground-based and airborne field campaigns to monitor ambient aerosol particles (e.g. Schwarz et al., 2006, 2010; Lamb et al., 2018, 2021; Katich et al., 2023). During a typical research flight in a source region, the SP2 might acquire signals for ∼4 million individual aerosol particles, necessitating automated detection techniques to process L-II signals from the instrument. Typical automated methods to process L-II signals use curve-fitting techniques and calibrations from laboratory studies to determine aerosol particle mass and coating thickness assuming a Mie-theory core shell

framework (e.g. Gao et al., 2007; Schwarz et al., 2006, 2010). As an alternative approach to traditional SP2 data processing techniques, Lamb (2019) explored a supervised machine learning approach.

Machine learning is increasingly being applied to observations from the specialized instrumentation used to detect *in situ* aerosol and cloud particles, providing new insights into the properties of atmospheric aerosol and cloud particle populations and facilitating the rapid analysis of large atmospheric data sets. While initial efforts have focused on supervised machine learning

(for example, for the Wide-band Integrated Bioaerosol Sensor (WIBS) (Ruske et al., 2017, 2018), the Particle Analysis By Laser Mass Spectrometry (PALMS) instrument (Christopoulos et al., 2018; Zawadowicz et al., 2017), and for cloud particle imager probe images (e.g. Przybylo et al., 2022)), more recent efforts have also explored unsupervised and self-supervised machine learning approaches, sometimes in combination with co-sampled environmental observations (e.g. Allwayin et al., 2024; Ko et al., 2025). As large datasets from *in situ* measurements are increasingly being compiled across field campaigns

and sampling conditions, these methods promise to lead to new physical insights into the atmospheric processes that underlie these observations (Lamb et al., 2025).

Supervised machine learning, involving the training of algorithms on a labeled dataset, has been used in the past to classify different types of absorbing aerosols with detectable incandescence in the SP2 (Lamb, 2019). While supervised machine learning methods can classify aerosols based on features derived from L-II signals, the algorithms need to first be trained on

labeled data sets. Lamb (2019) previously used observations of laboratory proxies for typical atmospheric aerosols detectable by the SP2 to create a labeled data set. While supervised machine learning performs well in classifying laboratory data sets by type, it has limited ability to generalize to ambient atmospheric aerosols, especially in cases where laboratory proxies are not readily available or when aerosol populations that have not been previously identified in past data sets are measured during airborne field campaigns.

On the other hand, unsupervised machine learning algorithms discern inherent patterns and correlations in data sets, without requiring predefined categories. These methods have not previously been applied to the problem of classifying L-II signals,

and they may address the limited generalization performance of supervised methods to ambient aerosol populations where laboratory proxies are not available or only partially replicate the characteristics of ambient aerosols. Here we explore how unsupervised machine learning can be applied to L-II signals from the SP2, with the goal of identifying different populations

of aerosols based on the information in their L-II signals alone. We also investigate whether unsupervised methods can provide insights into the variability of aerosols of different types based on their L-II response in the SP2. Our analysis provides insight into the amount of independent information that can be gained from L-II signals in terms of identifying the composition of refractory aerosols that reach detectable incandescence in the SP2, and also provides a starting point for future studies on how data-driven methods might provide new insights into the sources and variability of aerosol populations.

To demonstrate this method, we focus on the application of unsupervised machine learning to observations of laboratory proxies for several types of anthropogenic and natural aerosols that reach detectable incandescence in the SP2 (Lamb, 2019). We briefly describe these data sets, data pre-processing, and the unsupervised machine learning algorithms in Section 2. We then discuss the application of principal component analysis (PCA) and a variational auto-encoder (VAE) to the laboratory samples in Sections 3 and 4. In Section 5 we discuss how these approaches can be used to quantitatively analyze the similarity

of different aerosols detected by the SP2, and in Section 6 we explore how these methods can be used to improve classification of aerosols that the SP2 is sensitive to. Finally, in Section 7, we discuss the potential for machine learning to improve SP2 data analysis and interpretation.

## 2 Methods

### 2.1 Dataset

To investigate how effectively unsupervised machine learning can be used to differentiate different types of aerosols detected by the SP2, we use a labeled data set of L-II time series (Lamb, 2025) that was previously described in detail in Lamb (2019). This dataset is comprised of L-II signals obtained from measuring laboratory proxies for aerosols typically found in the atmosphere that reach detectable incandescence in the SP2. Each L-II signal represents an individual aerosol detection event in the SP2, and we treat each signal as an independent sample in applying the machine learning algorithms. The data set includes examples

of observations for 7 classes of aerosols: Fullerene Soot (FS), Fullerene Soot coated with glycerol (FS+glyc), Clifty Fly Ash (CFA), Arizona Test Dust (ATD), Volcanic Ash (VA), Iron (III) Oxide ($Fe_2O_3$), and Iron (II,III) Oxide ($Fe_3O_4$). The total number of aerosols of each class measured is given in Table 1.

This laboratory data set was developed to create a relatively balanced data set that includes examples of aerosols that the SP2 might observe in the atmosphere. In ambient conditions, the majority of aerosols that the SP2 observes are rBC particles.

Fullerene Soot is a laboratory proxy with a similar response in the SP2 as rBC. BC observed in the atmosphere is typically coated with non-absorbing materials, and we use the FS+glyc samples as examples of typical L-II signals of thinly coated rBC particles. Several studies have demonstrated that SP2s that have been modified to provide greater spectral contrast between their narrow and broadband detectors (such as the NOAA SP2 used in this study) can also detect iron oxide aerosols associated with anthropogenic combustion sources with high efficiency: magnetite ($Fe_3O_4$) can be detected with nearly 100% efficiency

**Table 1. Overview of laboratory data sets**.

| Class # | Class | Total | % with Detectable Incandescence |
|---|---|---|---|
| 0 | FS | 20004 | 98.28 |
| 1 | FS+glyc | 20018 | 81.04 |
| 2 | CFA | 20009 | 8.53 |
| 3 | ATD | 20001 | 23.93 |
| 4 | VA | 20005 | 27.00 |
| 5 | $Fe_2O_3$ | 20008 | 91.85 |
| 6 | $Fe_3O_4$ | 20037 | 98.21 |

under typical conditions, while the detection of hematite ($Fe_2O_3$) is lower and size-dependent (Yoshida et al., 2016). The two iron oxide powders ($Fe_2O_3$ and $Fe_3O_4$), have a similar response in the SP2 as anthropogenically-sourced iron oxide aerosols from combustion sources (Moteki et al., 2017; Lamb, 2019; Lamb et al., 2021), which we refer to as $FeO_x$ following past literature. In addition, the SP2 also detects measurable incandescence in a small fraction of aerosols with metallic inclusions, such as mineral dust, coal fly ash, and volcanic ash (Heimerl et al., 2012; Lamb, 2019). Here we use Arizona Test Dust as a

laboratory proxy for mineral dust, and the coal fly ash is Clifty-F. The volcanic ash was collected on the ground in Iceland from the Eyjafjallajökull Volcano.

## 2.2 Pre-processing L-II Time Series from the SP2

In this study, we focus on applying unsupervised machine learning to L-II signals from the SP2 instrument. In a previous study using supervised machine learning to classify aerosols detected by the SP2, significant feature engineering was used

to derive specific, interpretable features from the L-II time series (Lamb, 2019). By contrast, as input to our unsupervised machine learning algorithm, we use the unprocessed L-II signals, which are 80 $\mu$s time series consisting of 400 points (dt = 0.2 $\mu$s, see Figure 2 for examples), under the assumption that a PCA decomposition or alternatively, the non-linear deep learning method (VAE), will learn higher order features directly from the unprocessed L-II time series, that will provide insights into the variability and distinguishability of the observed aerosol particles.

For each of the 4 detection channels, we define a feature matrix $\mathbf{X_i}$, where $\mathbf{X_i} \in \mathbb{R}^{N \times t}$ is an $N \times t$ matrix. $N$ is the total number of L-II signals (the total number of observed aerosol particles), $t = 400$ corresponds to the number of time points in each signal, and $i \in [0, 3]$ corresponds to the detection channel in the SP2 instrument. The 0th channel corresponds to the scattering channel, the 1st channel corresponds to the "blue" incandescent channel, the 2nd channel corresponds to the "red" incandescent channel, and the 3rd channel corresponds to the position sensitive detector.

The typical approach to derive information from L-II signals from the SP2 is to find the maximum values for the scattering and incandescent channels, as these are proportional to the optical size and refractory mass of an aerosol, respectively. We define the maximum of channel 0 as $S_{max} = max(\mathbf{X_0})$, and the maximum of channel 1 as $I_{max} = max(\mathbf{X_1})$. In addition, the

"color temperature ratio", is defined as

$$CR = \frac{max(\mathbf{X_1})}{max(\mathbf{X_2})}, \tag{1}$$

the ratio between the peaks of the blue and red incandescent signals. CR is proportional to the blackbody temperature of the aerosol as it incandesces in the laser beam. Here the gains on the blue and red detectors have been chosen such that the CR $\approx 1$ for rBC (corresponding to a characteristic blackbody temperature of 4320 K) and CR $\approx 0.7$ for FeO$_x$ (corresponding to 3300 K). In practice, there is a significant amount of variability in CR across the population of aerosols of each class detected by the SP2. For further details of typical SP2 analysis, see discussion in Schwarz et al. (2006, 2010); Lamb (2019).

To first order, FeO$_x$ signals can be differentiated from rBC signals in the SP2 by differences between their blackbody temperature (CR) and their incandescent mass (proportional to the peak of the incandescent channel, $I_{max}$). However, CR and $I_{max}$ do not provide complete separation between the two classes, particularly when $I_{max}$ is small (i.e. for less massive particles) (Lamb, 2019). FeO$_x$ also demonstrates a less skewed incandescent peak in the SP2 than rBC signals, likely due to the metallic aerosols melting in the SP2 laser beam as they are heated to incandescence (Adachi et al., 2016; Lamb, 2019). In 145 addition, other types of aerosols detected by the SP2 such as ATD, VA, and CFA exhibit a broad range of $I_{max}$ and CR values, due to the presence of metallic inclusions with a variety of chemical compositions. The maximum value of the scattering channel, $S_{max}$, is proportional to the total optical size of the aerosol particle (except in cases when particles are large and the scattering channel is saturated). ATD, VA, and CFA generally also demonstrate significant scattering in the laser beam after the main incandescent peak, due to incomplete evaporation of these aerosols in the SP2. Because these particles are generally larger 150 than typical rBC particles, the scattering channel is more likely to be saturated for these aerosol types. Coated rBC and FeO$_x$ particles can be identified from the Ch. 0 time series due to an initial peak in the scattering signal as the coating evaporates from the particle, followed by a second peak when the refractory portion of the particle evaporates in the laser beam.

In this study we focus on particles that have detectable incandescence in the SP2. Therefore, we first remove any L-II signals where $I_{max}$ is close to the Ch. 1 signal baseline ($I_{max} < 0.2$). The majority of the rBC and FeO$_x$ aerosols have detectable 155 incandescence. However, only a fraction of the ATD, CFA, and VA aerosols demonstrate detectable incandescence in the SP2 (Table 1), likely because only a fraction of the particles in these aerosol populations have sufficient metallic inclusions (Lamb, 2019).

Pre-processing data is a typical first step for applying deep-learning algorithms to data sets, as methods work best when the input features are normalized between 0 and 1 and normally distributed. In our analysis, we tested several potential methods 160 for pre-processing Ch. 0 and Ch. 1 time series, which we delineate here:

1. *Division by the maximum of each channel across all samples in the training dataset.*

2. *Normalization of each channel by the minimum and maximum across all samples in the data set*

3. *Normalization of the channel by the minimum and maximum of each individual sample in the data set*

4. *Relative scaling for each sample using logarithmic normalization.*

165 We found that the third approach demonstrated the most promise in terms of separability of classes within the latent space learned by the VAE analysis that we describe in the rest of this paper. To pre-process the raw L-II signals for unsupervised machine learning, we therefore normalized Ch. 0 and Ch. 1 such that the time series for the scattering and blue incandescent channels are normalized between 0 and 1. That is, the input feature vector for our algorithm is

$$\mathbf{X_i^{scaled}} = \frac{\mathbf{X_i} - min(\mathbf{X_i})}{max(\mathbf{X_i}) - min(\mathbf{X_i})} \tag{2}$$

170 where $i \in [0, 1]$. The pre-processing method that we choose impacts the meaning of latent variables that are learned by the VAE and the principal components that are identified by the PCA analysis. In this case, the normalization approach that we have chosen means that we focus on learning compressed latent representations that describe the shape of the signals for Ch. 0 and Ch. 1, under the assumption that the shape of the L-II signal alone (without information about magnitude) can provide information that can be used to differentiate between different types of aerosol particles that are detected by the SP2.

## 2.3 Dimensionality Reduction of the L-II Signals

After pre-processing the raw L-II signals for Ch. 0 and Ch. 1, the data sets are randomly split into training, validation, and test data sets based on the aerosol sample number. We use 50% (85,832 samples) for training, and 25% (42,899) for validation and 25% (42,960) for testing our unsupervised machine learning approach.

To apply unsupervised machine learning, we apply both principal component analysis (PCA) and a variational auto-encoder 180 (VAE) to the L-II signals. PCA is a linear dimensionality reduction technique that decomposes signals into their principal components and corresponding weight vectors, where principal components correspond to the directions of greatest variance in the data, in decreasing order (Pearson, 1901). A VAE is a type of generative machine learning model that is designed to generate new data that is similar to the data that it is trained on (Kingma and Welling, 2022). It does this by learning a non-linear mapping from a higher-dimensional feature space to a lower dimensional latent space representation, which can then be 185 sampled to generate new data points. VAEs are widely used in machine learning for tasks like image generation and anomaly detection (Wei et al., 2020). In their ability to decompose signals into lower dimensional representations, VAE's are effectively non-linear counterparts to PCA.

A VAE consists of two neural networks with trainable weights, which are the encoder and the decoder models (Figure 2, bottom panel). The training of a VAE involves optimizing the parameters of both the encoder and decoder in order to 190 maximize the reconstruction of the original higher-dimensional input from its lower-dimensional latent space representation, while also ensuring that the learned latent space representation is smooth and continuous. The encoder takes an input signal and encodes it into a lower dimensional latent representation (Figure 2). The action of the encoder can be represented as a function, $q(z|x, \theta_{enc})$, where $x$ is the input signal, $z$ is its latent representation, and $\theta_{enc}$ are the weights of the encoder neural network. The encoder outputs parameters to a probability distribution, assumed to be Gaussian, which are the mean $\mu$ and 195 variance $\log(\sigma^2)$. Meanwhile, the decoder takes a point $z$ in the latent space and reconstructs the input $\hat{x}$. The decoder defines a probability distribution over the possible outputs given a latent point. For this reason, the decoder can be represented as

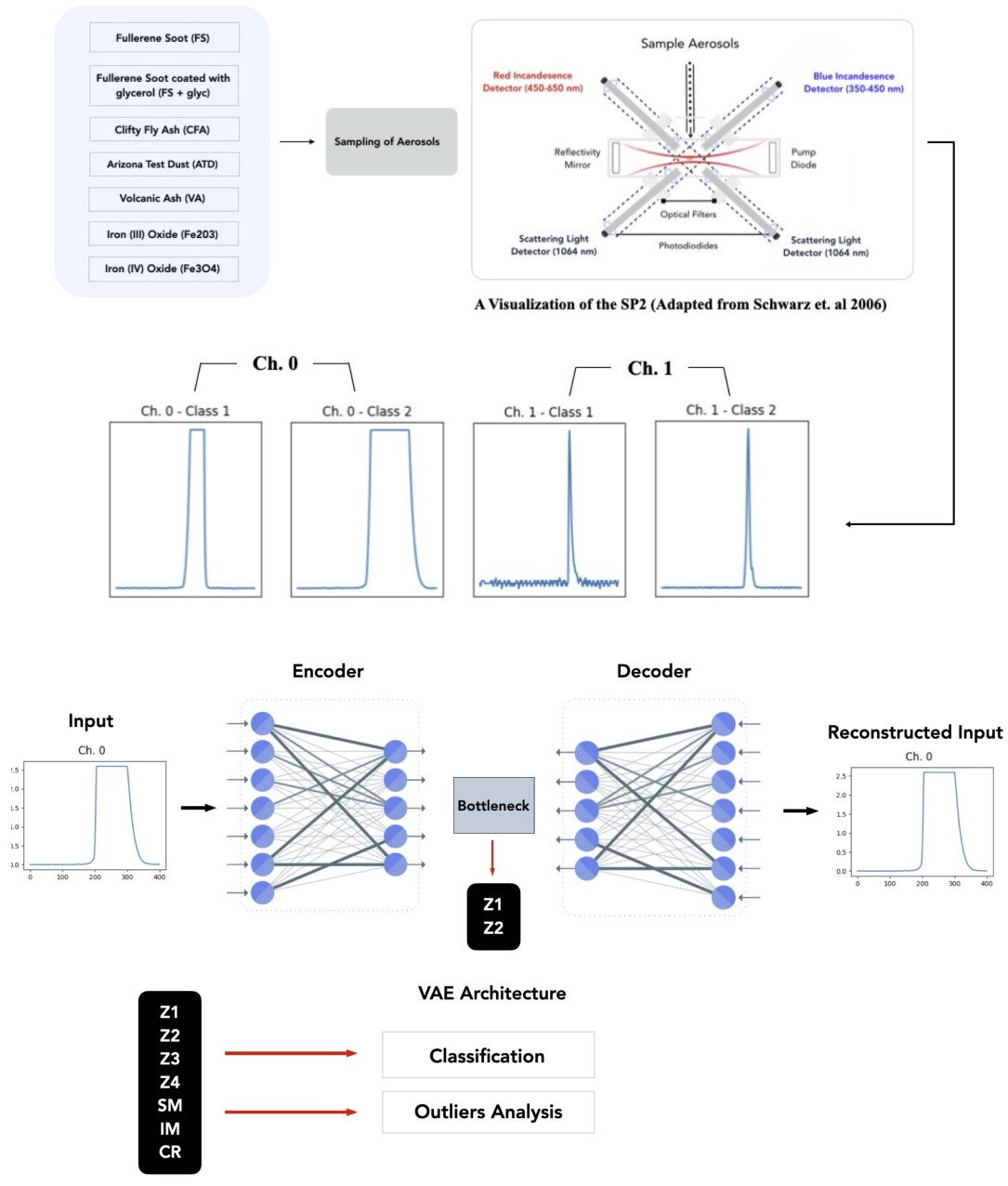

**Figure 2. An overview of the unsupervised machine learning approach applied to observations from the SP2.** First, aerosols are sampled by the SP2 and then after pre-processing, a variational autoencoder is trained to learn a lower dimensional latent representation of the L-II signals from each detection channel. We then explore several downstream tasks, using the latent space representations learned from the L-II signals. For comparison with the VAE, we also perform a PCA analysis of the L-II signals. The class # corresponds to Table 1.

$p(\hat{x}|z, \theta_{dec})$ where $\theta_{dec}$ are the weights of the decoder neural network, and $\hat{x}$ is the reconstructed signal (Kingma and Welling, 2022).

In order to learn the weights $\theta_{enc}$ and $\theta_{dec}$ of the encoder and decoder networks, we minimize the loss function for a VAE, which is the sum of the reconstruction loss and the similarity loss (Kingma and Welling, 2022). The reconstruction loss ensures that the decoded samples match the original inputs, while the similarity loss ensures that the learned latent representation is smoothly varying.

For reconstruction loss, we use binary cross-entropy loss.

$$L_{\text{reconstruction}} = -q(z|x) \left[ \log p(x|z) \right] \tag{3}$$

We also explored using mean squared error loss for the reconstruction loss and found that this did not make a significant difference in our analysis of the L-II signals.

The similarity loss ensures that the distribution of latent variables ($z$) stays close to a prior distribution, which is assumed to be a standard normal distribution (Kingma and Welling, 2022). This term acts as a regularizer and effectively ensures that the learned latent space is smoothly varying:

$$L_{\text{similarity}} = D_{\text{KL}}(q(z|x) \| p(z)), \tag{4}$$

where $D_{\text{KL}}$ is the Kullback-Leibler divergence between the encoder's distribution $q(z|x)$ and the prior distribution $p(z)$. The KL-divergence measures the distance between two data distributions, and is defined as,

$$D_{\text{KL}}(p(x) \| q(x)) = \int -p(x) \ln \left( \frac{p(x)}{q(x)} \right) dx \tag{5}$$

The total loss $L_{total}$ for the VAE can then be computed by summing the reconstruction loss and similarity loss:

$$L_{\text{total}} = L_{\text{reconstruction}} + L_{\text{KL}} \tag{6}$$

Here we use the VAE algorithm implemented in the pyroVED library (Ziatdinov; Biswas et al., 2023), which is built on top of the Pytorch deep learning library and the Pyro probabilistic programming language (Bingham et al., 2018). The pyroVED library minimizes Eq. 6 using stochastic variational inference, using the Adam optimizer (Kingma and Ba, 2017).

For the L-II signals from the SP2, we independently train two VAE's for 200 epochs each on the normalized Ch. 0 and Ch. 1 signals, respectively. Here, we are interested in extracting information about the shape of the L-II signals from the scattering and incandescent channels, under the assumption that the shape of the signals can provide information about the type and characteristics of the aerosol that was measured in the SP2. The latent variables $z$ effectively characterize the shape of the normalized L-II signals, and we train two VAEs in order to independently learn representations for the normalized scattering and incandescent channels.

Since the specific latent representations are learned from the distribution of data that the VAE is trained on, the VAE in general will not learn the same latent representation from a new training data set. Training on a representative sample of

laboratory measurements (as we do here) can provide a wide range of possible samples for the generative model. However, once trained, the VAE can be applied in its inference mode to any new data set, providing the same mapping between input signals and lower dimensional representations. For example, the VAE could be trained on laboratory samples and then can be applied to ambient measurements, allowing a comparison between the latent representations of ambient aerosols and laboratory proxies.

## 3   Principal Component Analysis of the L-II Signals

To provide a basis for comparison with the non-linear deep learning approach (the VAE we describe in the next section), we first apply PCA to the L-II signals. This allows us to quantify the potential value added by the non-linear dimensionality reduction provided by the VAE. PCA also has the advantage of being more directly interpretable than the VAE, and thus in some cases might be preferable to use for the SP2 analysis, as there is a direct correspondence between the identified PC's, their corresponding weights, and the decomposition of the signals. Here, we use the scikit-learn library implementation of PCA(Pedregosa et al., 2011), and apply the algorithm separately to the pre-processed signals for Ch. 0 and Ch. 1.

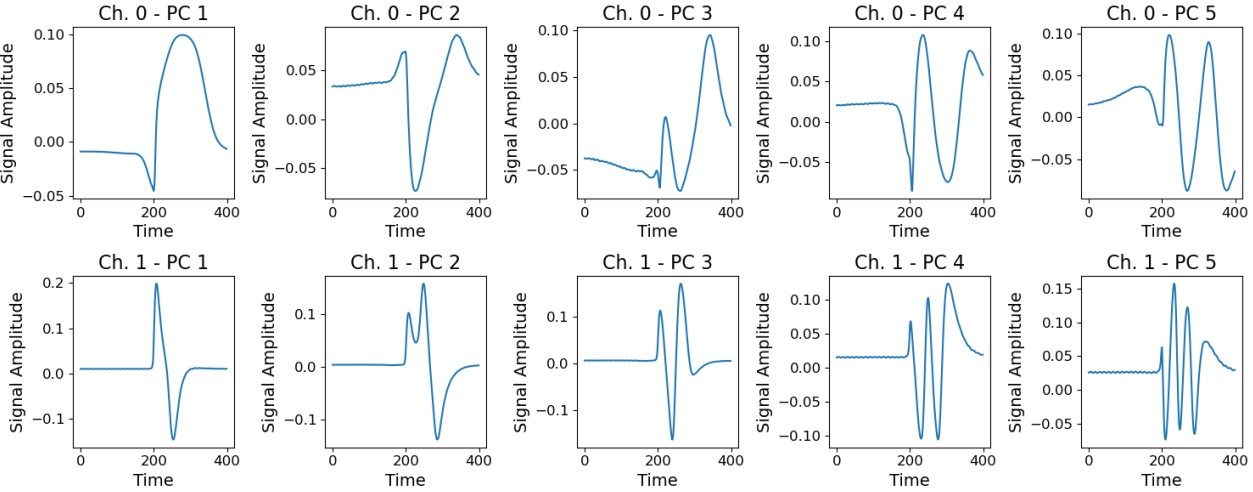

**Figure 3. First 5 Principal Components of the L-II signals** Top row: Ch. 0 (Scattering channel). Bottom row: Ch. 1 (Incandescent channel)

We first investigate how many principal components (PCs) we need to keep to explain 95% of the variance in the L-II signals. We find that Ch.0 (the scattering channel) requires 11 PC's and Ch.1 (the incandescent channel) requires 18 PC's to account for 95% of the variance across our training data set. In the case of Ch. 0, the first 2 PC's account for 56.9% and 17.9% of the variance, whereas the first two PC's for Ch. 1 account for only 30.0% and 17.7% of the variance, respectively. We visualize the first 5 PC's for Ch. 0 and Ch. 1 in Figure 3.

We next visualize the weights associated with the first two principle components for Ch. 0 and for Ch. 1, and investigate the differentiability of the different classes in this space (Figure 4). By examining the separability of the different aerosol classes in

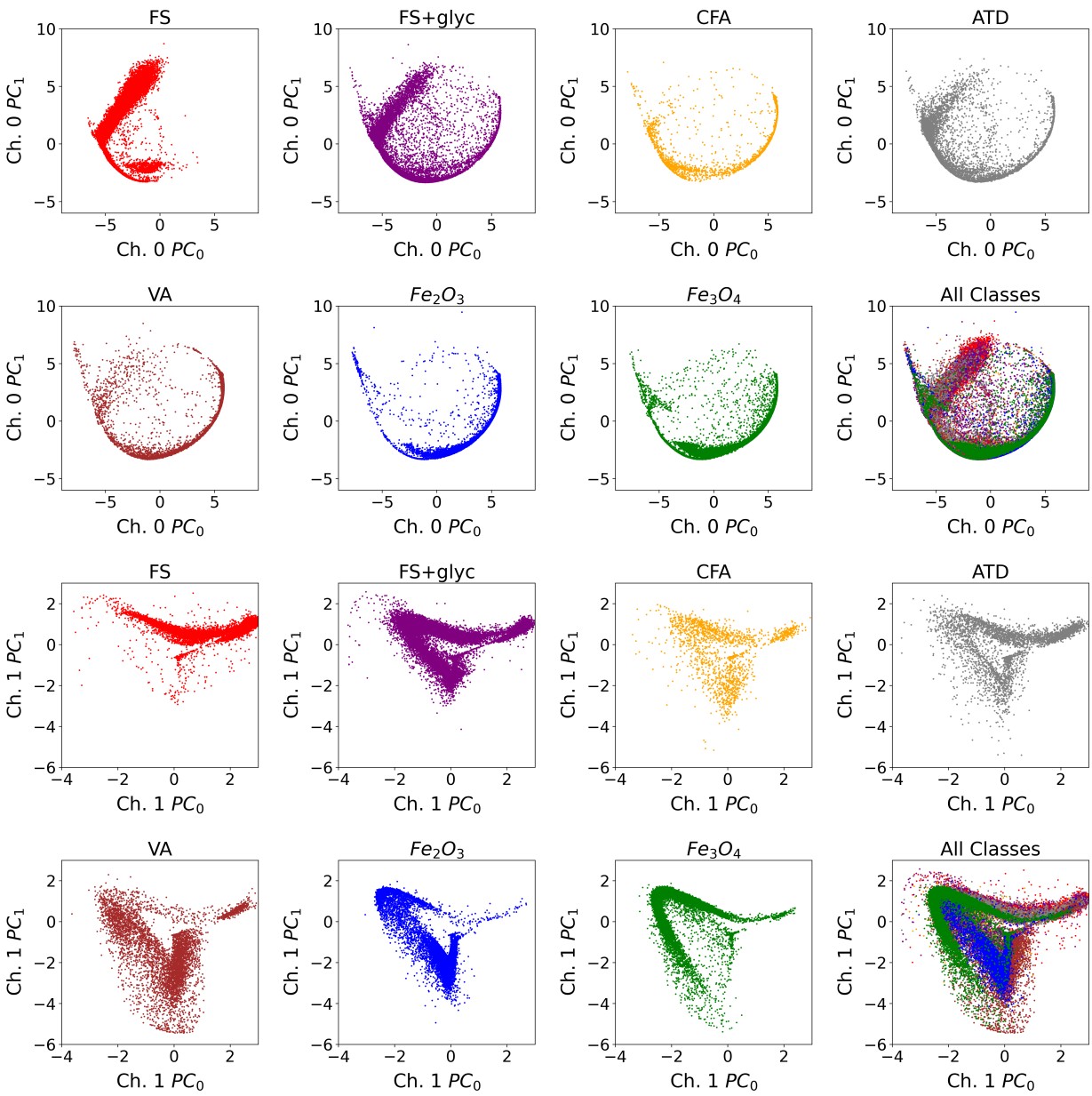

**Figure 4. Weights associated with the first two principal components by aerosol class** Each point represents a single L-II time series associated with an aerosol detection in the SP2, and we color the points by aerosol class. The location of each point corresponds to the value of the weights associated with the first two principal components for Ch. 0 (Ch. 0 $PC_0$ and Ch. 0 $PC_1$, top panels) and Ch. 1 (Ch. 1 $PC_0$ and Ch. 1 $PC_1$, bottom panels).

this space, we do see that there are some differences between aerosol class. For example, while many of the weights associated with the first two PCs of Ch. 0 lie along a curve in this space, FS and FS+glyc show a distinctive linear branch in the top left side of the plot, which is not evident for the iron oxide aerosols. In addition, FS+glyc has many more examples of aerosols that lie along the lower left triangle of the manifold that most aerosols occupy in Ch. 1 $PC_0$ vs. Ch. 1 $PC_1$ space when compared with FS. Similarly, the two iron oxide aerosols demonstrate differences in terms of the highest density of points along the lower left side of the manifold, and the location of these dense points are distinctly separated in this space. This suggests that the weights of the first two PCs of the signals could provide information from the signals that might help to separate aerosols of different types in terms of their response in the SP2.

## 4  Analysis of Learned Latent Representations of the L-II signals

We next use the trained VAEs to encode the L-II signals from the training data set for both Ch.0 and Ch.1 into lower dimensional latent representations, which we refer to as $z_i$. We refer to the latent variables for Ch. 0 as $z_1$ and $z_2$, and the latent variables for Ch. 1 as $z_3$ and $z_4$. Because these latent variables are learned representations of the normalized Ch.0 and Ch.1 time series, these variables provide information about the shape of these signals. Thus, the distance between variables in the latent space representations provide a means to visualize how similar L-II signals are to one another.

The smoothness constraint in the VAE ensures that signals with greater similarity are mapped to points that are closer together in the latent space, preserving meaningful structure in the learned representation. This constraint encourages continuity, meaning that small changes in the input signal result in gradual variations in the latent representation. However, the specific structure of these learned representations is inherently shaped by the underlying data distribution, as the VAE optimizes its encoding based on the patterns present in the underlying data. The distribution of latent variables for each aerosol class can provide information about how similar signals within each class are to one another, and it can also provide information about how much separability there is between different aerosol classes in terms of their latent space representations. These distributions provide insights into whether the shape of the normalized signals for Ch. 0 and Ch. 1 can be used to differentiate classes of absorbing aerosols that the SP2 is sensitive to.

The latent variables learned from the normalized Ch. 0 and Ch. 1 signals demonstrate smoothly varying characteristics when we plot the distributions of the aerosol populations in terms of $I_{max}$ vs. CR (Figure 5). $I_{max}$ (proportional to the mass of the refractory portion of the aerosol) and CR (proportional to the temperature at which an aerosol incandesces in the SP2's laser beam) are strongly correlated with both the size of the aerosol and its chemical composition. As discussed in Section 2.2, the left mode in Figure 5 is typical of $FeO_x$ aerosols, and the right mode is typical of rBC aerosols and their proxies, including FS, due to differences in their characteristic incandescent temperatures. Since the latent representations are smoothly varying in this space, this suggests that the shapes of both the scattering and incandescent signals are strongly correlated with their overall distributions in terms of refractory aerosol mass and incandescent temperature. This suggests that the shape of the L-II signals alone (without information about the signal magnitude) provides information about the physio-chemical properties of the aerosols detected in the SP2. We see a similar smoothly varying relationships between the weights associated with the

principal components for Ch. 0 and Ch. 1 and the refractory mass vs. incandescent temperature (not shown), further indicating
that the shape of the L-II signals alone provide significant information about the properties of the aerosols.

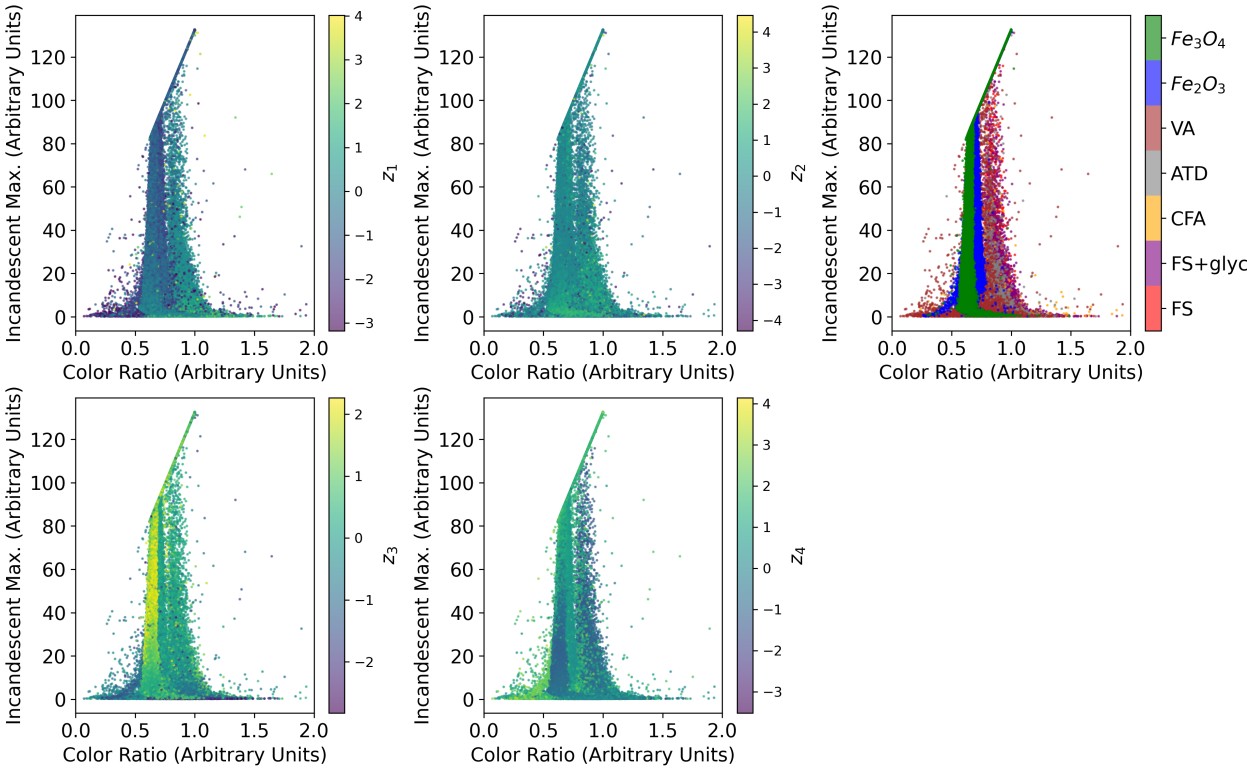

**Figure 5. Incandescent-peak-height to color-ratio relationship for different incandescent aerosols visualized by latent representation:** Each point represents a single aerosol particle, and we color the points by the latent space representations for Ch. 0 ($z_1$ and $z_2$, top panels) and Ch. 1 ($z_3$ and $z_4$, bottom panels). The aerosol class is shown in the top left panel.

To better understand how the latent variables represent the shapes of the L-II signals, we use the trained decoders from each VAE to map out the latent space representations in terms of representative Ch. 0 and Ch. 1 time series. Figure 6 shows the latent manifold for Ch. 0, left, and for Ch. 1, right. By generating characteristic signals along the deciles of a gaussian distribution of the latent variables for Ch. 0 and Ch. 1, we can examine how the latent variables capture specific properties of the time series signals such as symmetry and saturation. The saturation of signals in the Ch. 0 time series (Figure 6, left panel) shows flatness at the peaks, which is most common for dust-like particles (ATD, VA, and CFA), as the scattering detector becomes saturated for these large particles, and the peaks are artificially flattened. These signals are more evident in the bottom right side of the latent manifold for Ch. 0. The iron oxide aerosols $Fe_2O_3$ and $Fe_3O_4$ commonly have symmetric scattering signals, which is evident in the upper right of the latent manifold for Ch. 0. Because the signals are normalized between 0 and 1, the top left part of the latent space for Ch. 0 appears to pick up relatively small scattering signals, where the noise on the baseline is comparable

to the height of the peaks in the signal. The latent manifold for Ch. 1 captures the symmetry of the incandescent signal along one dimension, and the narrowness of the incandescent peak along the other dimension.

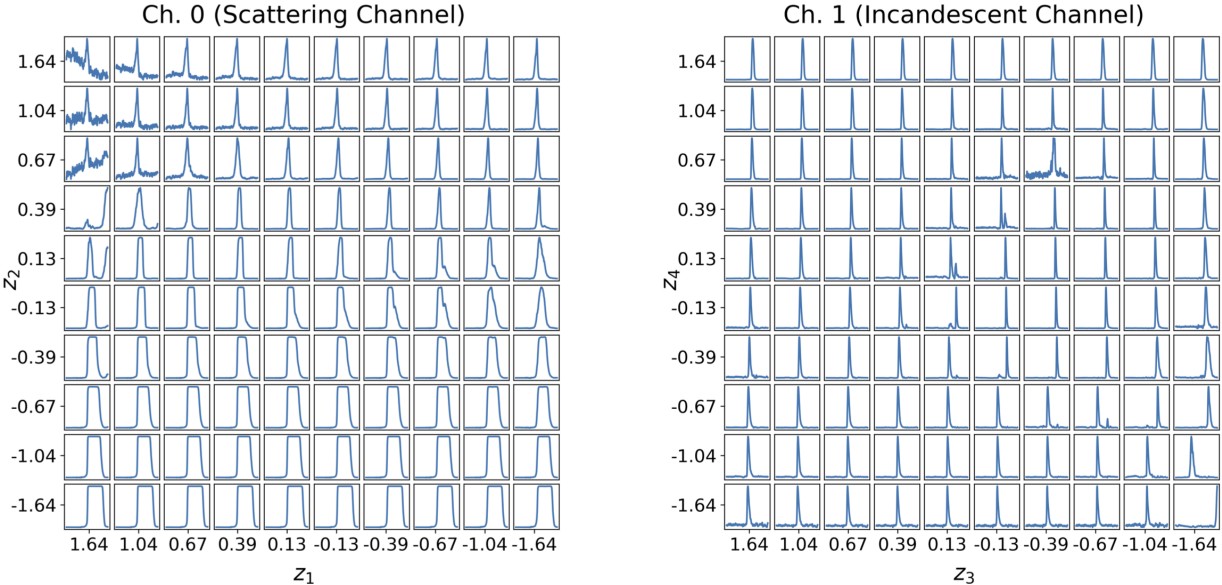

**Figure 6. Latent Manifolds for the L-II Signals.** By sampling along the deciles of a gaussian distributions for the 2 latent variables from each channel, we visualize characteristic L-II signals using the trained decoder. These learned latent manifolds provide an overview of the modes of variability described by the latent representations of the two channels. **Left:** Latent Manifold for Channel 0, showing the variability represented by $z_1$ (x-axis) and $z_2$ (y-axis). **Right**: Latent Manifold for Channel 1, showing the variability represented by $z_3$ (x-axis) and $z_4$ (y-axis).

To more quantitatively assess the types of variability learned by the VAEs for Ch. 0 and Ch. 1, we analyze the decoded signals associated with each decile, using the scipy package (Virtanen et al., 2020). We use the find_peaks function in scipy to find the largest peak in each time series, and then determine the peak location (in terms of digitized time points). We also use the peak_widths function to determine the width of the full width half maximum (FWHM) of this peak, as well as the digitized time point of the leading edge of the peak at half maximum (which we refer to as leading edge location). The peak locations, leading edge location, and FWHM associated with the decoded signals of the latent manifolds for both Ch. 0 and Ch. 1 are shown in Figure 7. For Ch. 0 in particular it is clear that the both the leading edge location and the FWHM are strongly correlated with $z_2$. For Ch. 1, the peak location and leading edge location have their highest values (associated with later times in the L-II signals) towards the center of the manifold. The signals at the center of the manifold are also associated with more narrow peaks in terms of their FWHM.

Since the size of the latent representations learned by the VAE is a hyper-parameter, we also explore how varying the size of the latent space impacts our ability to differentiate aerosols by class using these lower dimensional latent representations. To do this, we train VAEs with a latent vector $z$ that has either n=2 or n=3 variables. We then visualize the distributions of the latent

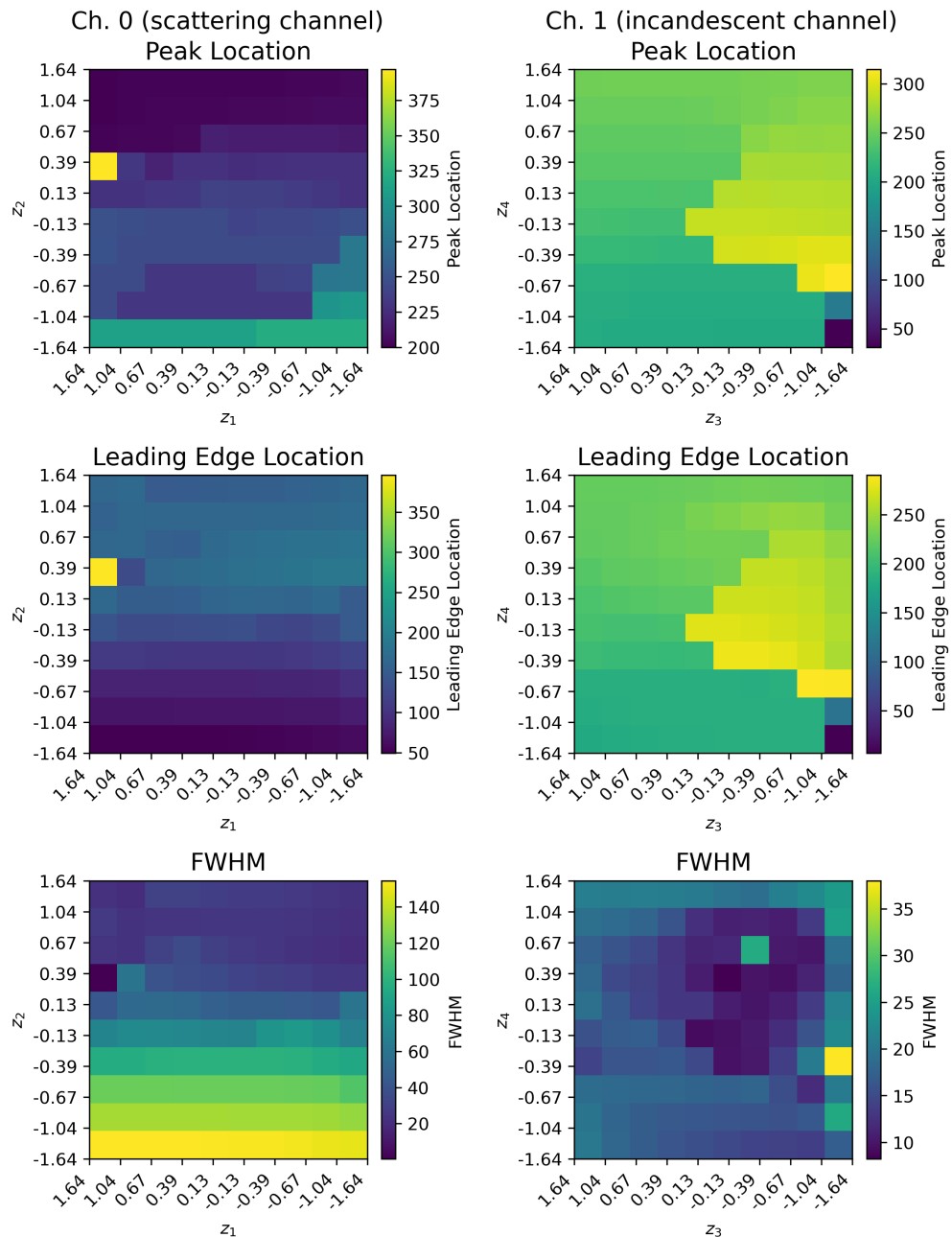

**Figure 7. Signal Analysis of Latent Manifolds for the L-II Signals.** The main peak location, peak leading edge location (defined as the leading edge of the largest peak at half maximum), and full-width half maximum for the decoded L-II signals shown in Figure 6 for Ch. 0 (left column) and Ch. 1 (right column). The peak location, leading edge location, and FWHM width are all given in terms of digitalized time points (out of 400 for the NOAA SP2).

variables from our training data set for each of the 7 aerosol classes represented in our dataset. Figure 8 shows the distribution of the encodings of Ch.0 (top) and Ch. 1 (bottom) when training two VAEs with n=2 variables in its latent space representation, and Figure 9 shows the distributions of the encodings of Ch. 0 (top) and Ch. 1 (bottom) when training two VAEs with n=3 variables. Each point on these plots represents the L-II signal from an aerosol detected in the SP2, encoded by the VAE into its latent space representation. These latent space representations therefore gives us a useful way to visualize the distributions of L-II signals found in each class.

In examining the latent space distributions for n=2 (Figure 8), we can find some consistency in terms of the encodings for different aerosol classes. For the scattering channel (Ch. 0) and incandescent channel (Ch. 1), the latent space representations for the black carbon proxies (FS and FS+glyc) show significant overlap in some locations, as do the latent space encodings for the FeO$_x$ proxies (Fe$_2$O$_3$ and Fe$_3$O$_4$). However, there are also regions of high density for the Ch. 1 embeddings of the FS+glyc samples that are not occupied by the FS samples. Similarly, for the two iron oxides, Fe$_2$O$_3$ has a higher density of points near the origin and for positive values of $z_4$ when compared to Fe$_3$O$_4$. This suggests that while there are some similarities between the latent embeddings for the two types of FS and for the two iron oxide aerosols, there are some notable differences between these classes in terms of their response in the SP2. We quantify this similarity further in Section 5. Regions of high density in the latent space representations indicate more examples of that L-II signal is found when measuring a specific aerosol class. The latent space representations for the dust-like aerosols (CFA, ATD, and VA) show less clear regions of high density when compared to the black carbon proxies and FeO$_x$ proxies, which is explained by the greater variability across classes in observed L-II signals for the dust-like aerosols. In Lamb (2019), it was noted that these dust-like aerosols were more likely to lead to saturated signals (due to their large optical size) or other irregularities in their L-II signals when compared with rBC or FeO$_x$. This greater variability in L-II signals is reflected in the greater spread in their latent space representations.

Similarly, the latent space distributions for n=3 (Figure 9) show some consistency across aerosols of similar classes. However, the additional dimension in the latent space representations provides additional contrast between aerosols of similar classes. For example, for FS+glyc, the latent space representation for Ch.1 in part significantly overlaps with that for uncoated FS, but also shows a large density of points in space that is not represented at all by the FS signals. In addition, while VA, ATD, and CFA show similarities in their latent space representations for Ch. 0, there are more clear differences in the density of points observed in their latent space representations for Ch. 1, suggesting that the incandescent signal provides greater contrast in differentiating the signals of these dust-like aerosols by class. Physically, this makes sense, since the L-II incandescent signal could provide greater contrast between metallic inclusions of different chemical composition or size than the scattering signal, and these likely differ between the ATD, VA, and CFA populations. Finally, the iron oxide aerosols also show more clear separability in terms of their latent space representations for Ch. 1.

We found that a latent space with n=2 or n=3 provided sufficient separability and interpretability of latent space variables of the L-II time series, compared to higher dimensional latent space representations. Increasing the size of the latent space enables the VAE to capture a higher level of detail and complexity in the original L-II signals, allowing for more accurate and nuanced reconstructions. However, high dimensionality for the latent space representation is more difficult to interpret, and may not provide additional meaningful information for downstream tasks. We explored a latent space with n=4 and n=5 (not shown),

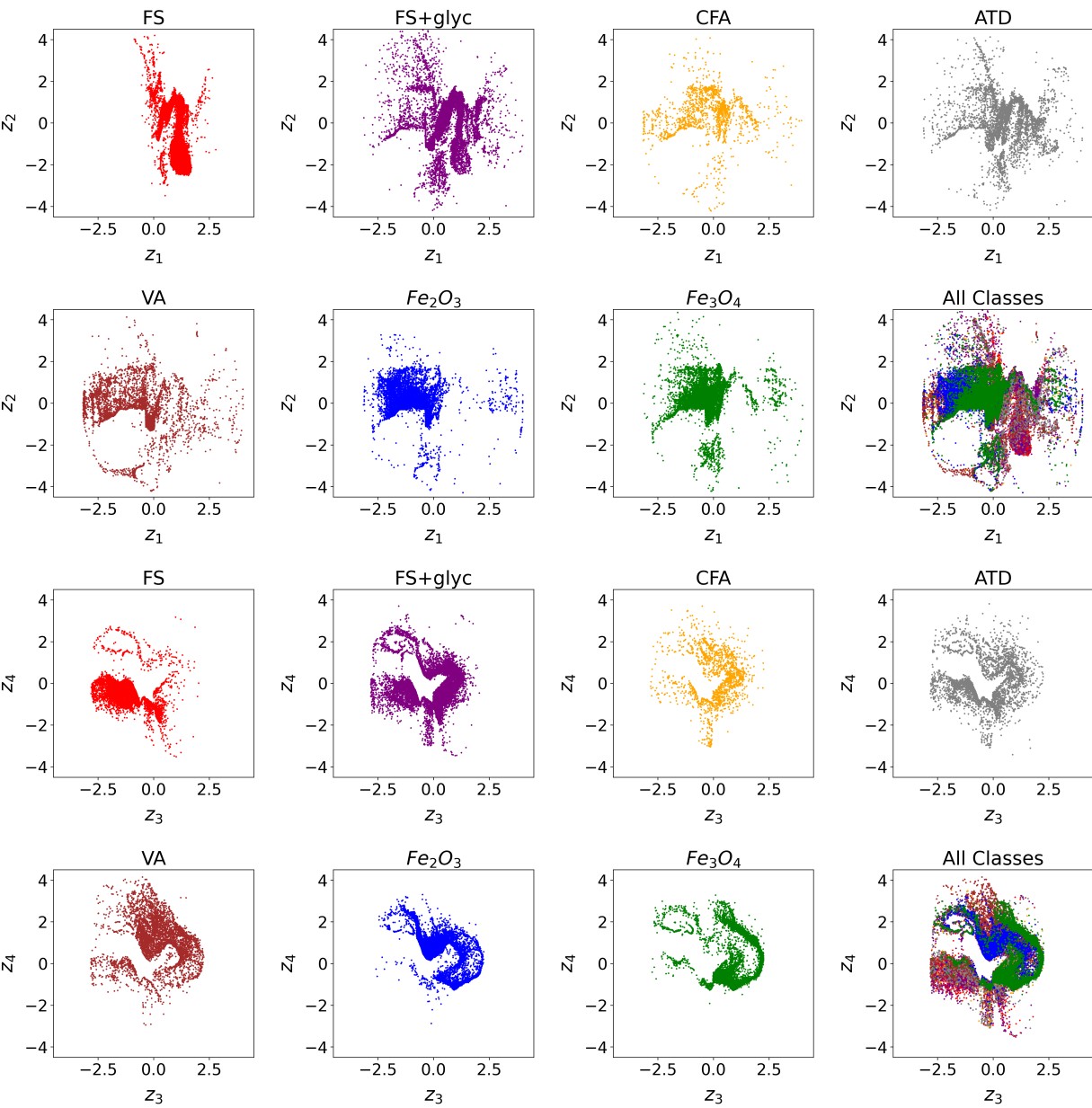

**Figure 8. Latent Space for n = 2.** Top 2 rows: Latent space distribution for Ch. 0 (scattering channel) shown by each aerosol class. Bottom 2 rows: Latent space distribution for Ch 1 (incandescent channel) shown by each aerosol class.

but found that including additional variables did not provide improved performance on downstream tasks like classification (Section 6), suggesting that the majority of the variance in the L-II signals can be encoded into 3 latent dimensions. We

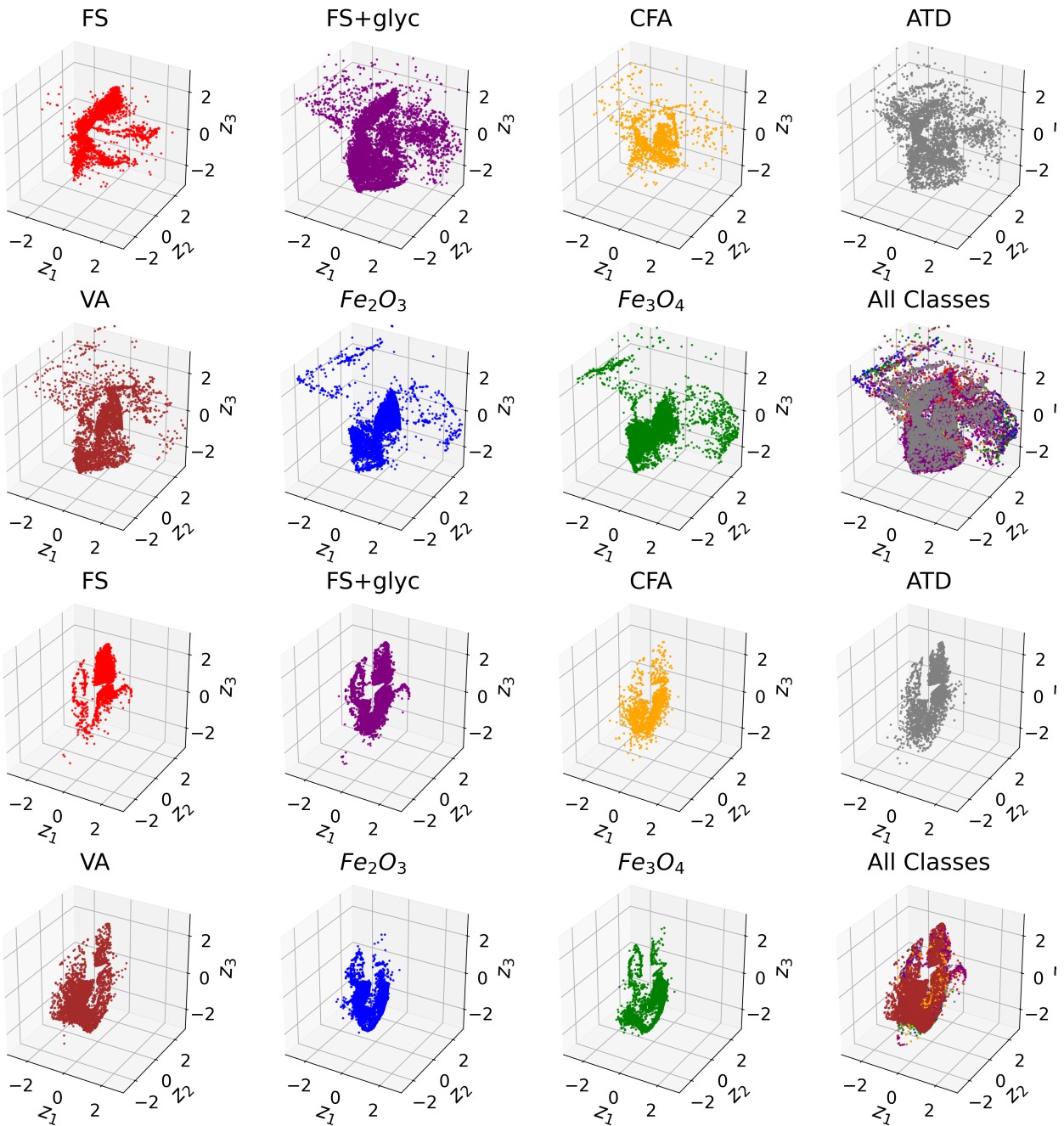

**Figure 9. Latent Space for n = 3.** Above: Latent space distribution for Ch. 0 (scattering channel) shown by each aerosol class. Below: Latent space distribution for Ch 1 (incandescent channel) shown by each aerosol class.

additionally estimated the intrinsic dimension of the manifold of the latent variables associated with Ch. 0 and Ch. 1 for n=2,3,4, and 5 latent variables. The intrinsic dimension is defined as the minimum number of variables required to represent the variability in a dataset. We used the intrinsic dimension algorithms in the scikit-dimension package (Bac et al., 2021), and found that independent of the intrinsic dimension algorithm used, the estimated intrinsic dimension was approximately 3 for the L-II time series for both channels.

## 5 Quantifying Variability in Aerosol Populations using the L-II Embeddings

One potential application of the dimensionality reduction methods that we have described in the previous two sections is to use the lower dimensional representations of the L-II signals to quantify the variability in ambient populations in terms of their responses in the SP2. We use cosine similarity as a metric to assess how similar the SP2 response to aerosols within each class are to each other and to the SP2 response to aerosols of different classes. The cosine similarity is defined as

$$cos(\theta) = \frac{X \cdot Y}{(||X|| * ||Y||)} \tag{7}$$

where X and Y represent the embeddings of two L-II signals. By embeddings we refer to either the first two PCs for Ch. 0 and Ch. 1 from the PCA analysis, or the latent variables from the 2D and 3D VAEs. The cosine similarity determines similarity by the normalized dot product of X and Y, where X and Y are vectors of length 2 (for PCA and the 2D VAE) or 3 (for the 3D VAE), and $\theta$ is the angle between the two vectors. This metric can have a value between -1 and 1, with higher positive magnitude indicating that vectors are more closely aligned in terms of direction and magnitude. Negative values indicate that the embeddings are not similar to one another.

To evaluate the within class and between class similarity for the 2D VAE, 3D VAE, and PCA embeddings of the L-II signals, we use the scikit-learn implementation of cosine similarity to determine the cosine similarity between all of the embeddings associated with samples in class $i$ and all of the embeddings associated with samples in class $j$. The diagonal represents when $i = j$, which is calculated by determining the mean cosine similarity for pairs of samples within the same class. The mean value of the pairwise cosine similarity for the embeddings from the various dimensionality reduction methods are displayed in Figure 10.

Although there are some slight differences between the results from the 2D VAE, 3D VAE, and PCA analysis, overall there are strong correlations between the mean cosine similarity values for Ch. 0 and Ch. 1 in each case. This suggests that irregardless of the dimensionality reduction method that we use (linear or non-linear) there is consistency in terms of the underlying data distributions and suggests a meaningful correlation with the known physio-chemical characteristics of the aerosols that are sampled. The values for mean cosine similarity for the 2D VAE and 3D VAE are generally slightly higher than the PCA analysis, which indicates the added value of using the non-linear dimensionality reduction, where higher-order latent features are determined from the distribution of the data itself.

The aerosols that are more optimally detected by the SP2 (FS, $Fe_3O_4$, $Fe_2O_3$) show higher within-class mean cosine similarity values. This indicates that the L-II signals associated with these classes are more similar to one another. The laboratory

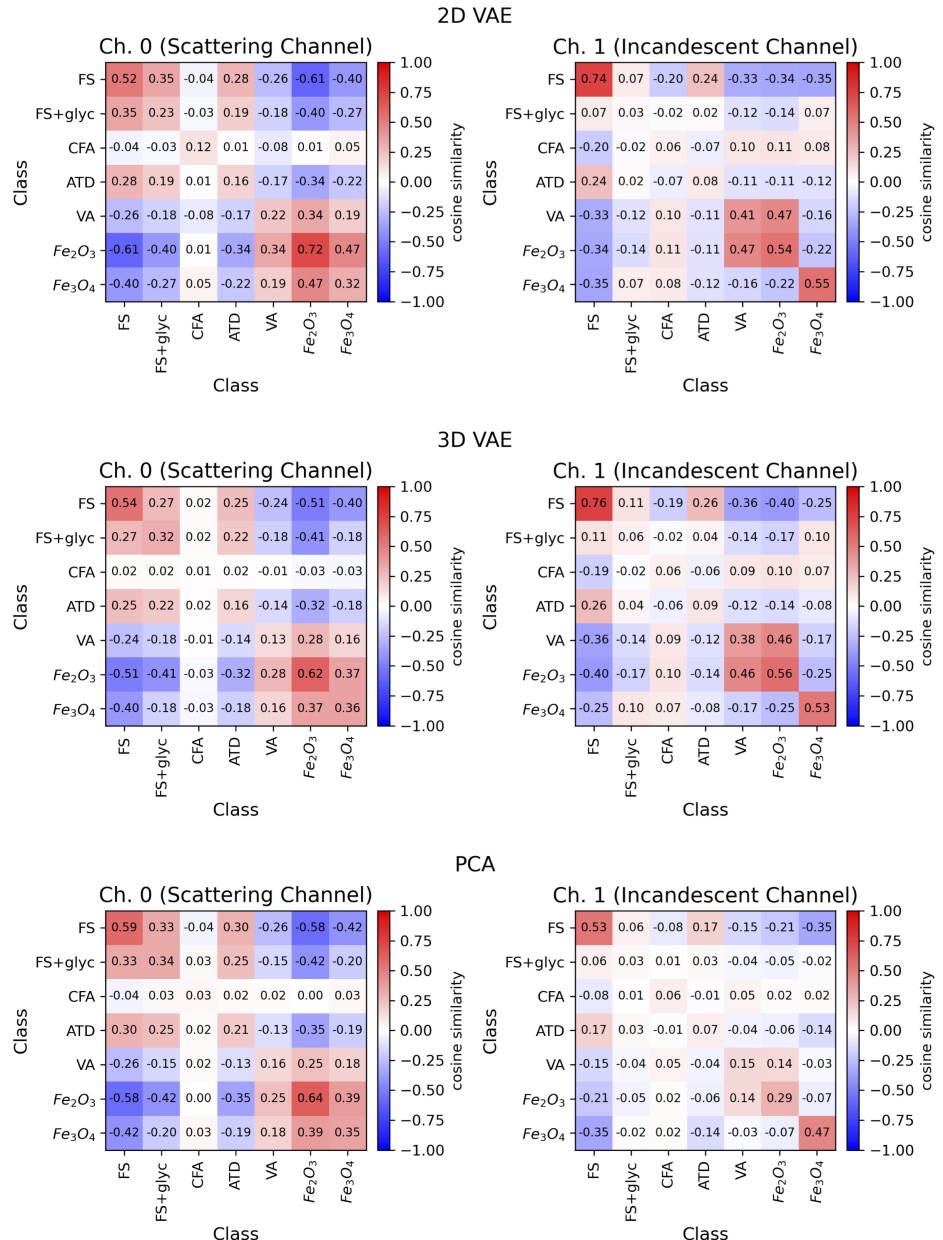

**Figure 10. Cosine Similarity of Embeddings for Aerosol Classes.** Top panels: 2D VAE, Middle panels: 3D VAE, Bottom panels: PCA

proxies for these aerosols are also likely more homogeneous than the dust-like aerosols (CFA, ATD, and VA). The dust-like aerosols, which the SP2 in general is not optimized to detect, show much lower mean cosine similarity scores within each class. These aerosols are composed of more heterogeneous materials, since dust-like aerosols that incandesce in the SP2 include metallic inclusions and are not pure samples with a known composition.

The pairwise cosine similarity metric also provides an opportunity to compare how similar L-II signals are between different aerosol classes. For the scattering signals (Ch. 0), FS and FS+glyc are more similar to one another, with ATD also being more similar to FS and FS+glyc. $Fe_3O_4$, $Fe_2O_3$, and VA also have higher between-class similarity for Ch. 0. The incandescent channel (Ch. 1) demonstrates slightly different relationships between the various classes of aerosols, with the $Fe_2O_3$, and VA showing stronger similarity, and FS and FS+glyc also showing some similarity. Because there are differences between the Ch. 0 and Ch. 1 mean pairwise similarity, this also demonstrates the added value of using information from both the scattering and incandescent channels to identify aerosols by class from their L-II signals.

## 6 Classifying aerosols using their L-II latent space representations

As a final down-stream task, we investigate how useful the lower dimensional representations of the L-II time series are in terms of differentiating absorbing aerosols that the SP2 is sensitive to by class. To do this, we follow an approach similar to the supervised classification approach, using a random forest algorithm to classify aerosols in a supervised manner, as described in Lamb (2019). However, rather than doing significant feature engineering on the L-II signals as input features to train the random forest, we instead use the first two PCs of Ch.0 and Ch. 1 (for the PCA) or the learned latent space representations from Ch. 0 and Ch. 1 (for the 2D and 3D VAE's).

For each sampled aerosol, we concatenate the lower dimensional embeddings of Ch. 0 and Ch. 1, as well as the $S_{max}$, $I_{max}$, and CR derived from the L-II signal for that aerosol (Figure 2). We test using the first 2 PCs from both Ch. 0 and Ch. 1, the 2 latent variables each from Ch. 0 and Ch.1, and the 3 latent variables each from Ch. 0 and Ch. 1. In the follow text, we refer to the case with the 2 latent variables each from Ch. 0 and Ch. 1 as the "2D latent space RF" and the 3 latent variables each from Ch. 0 and Ch. 1 as the "3D latent space RF". We refer to the case where we use the first two PCs from Ch. 0 and Ch. 1 as the "PC features RF"

The Random Forest algorithm constructs an ensemble of decision trees, using a subset of the training samples to construct each decision tree. The class for each sample is then determined by the vote of all the randomly constructed decision trees. Here, we use 50% of the data for training, 25% for validation, and 25% for testing the random forest, and use the Random Forest algorithm as implemented in the scikit-learn package (Pedregosa et al., 2011).

The confusion matrices for the results of the random forest classification for the 2D latent space RF (top left) and the 3D latent space RF (top right) and the PC features RF (bottom) is shown in Figure 11. All three RFs demonstrated similar performance. We also investigated 4D and 5D latent space representations as input to the RF, and did not find significant further improvements over the 2D or 3D cases, suggesting that 2 or 3 latent variables already provides the majority of meaningful information from the L-II signals in terms of differentiating absorbing aerosols.

Compared with the random forest that used significant feature engineering (Lamb, 2019), the 2D and 3D latent space RF's and the PC features RF are able to more accurately classify aerosols of each of the 3 major sub-types that the SP2 detects (BC, dust-like, and $FeO_x$). In particular, we find that the PC, the 2D, and the 3D latent space RF's are able to do a significantly better job of differentiating $Fe_2O_3$ and $Fe_3O_4$ from one another; in Lamb (2019) the RF with significant feature engineering

mis-identified $Fe_2O_3$ as $Fe_3O_4$ 50% of the time. This is likely due to the differences that are evident in the latent space representations from Ch. 1 (Figure 8), indicating that there are differences in the incandescent signals for $Fe_2O_3$ and $Fe_3O_4$ that are not readily evident from a simple analysis of the CR. These differences even showed up in the simpler linear PCA of Ch. 1, as can be seen in the differences in the PC weights for Ch. 1 $PC_0$ and Ch. 1 $PC_1$ in Figure 4. This suggests that there are meaningful differences between iron oxide aerosols in terms of their L-II signals, which could be further exploited to improve the detection accuracy of these aerosols using the SP2, and warrants further investigation. This result further illustrates how these dimensionality reduction techniques can identify new information from the L-II signals that is not obvious following traditional SP2 analysis approaches. The PC features, the 2D, and the 3D latent space RFs also do a slightly better job at differentiating the 3 different classes of dust-like aerosols from one another, particularly for VA, compared with the RF with significant feature engineering described in Lamb (2019). Overall, the 3D latent space RF performs slightly better than the 2D latent space and PC features RFs, likely due to the greater amount of information provided to the RF.

## 7 Conclusions

In this paper, we have built on our previous research in Lamb (2019) to further explore how data-driven methods such as machine learning can be applied to L-II time series from the SP2 instrument. Here we have focused on unsupervised machine learning as a method to classify aerosols with different chemio-physical properties, thus demonstrating a path towards better understanding the variability of aerosols observed in the atmosphere. By using both principal component analysis and a variational autoencoder to encode the L-II signals from the SP2, we demonstrated that lower dimensional representations of the L-II signals can be used to quantify the variability within and between aerosol classes. It can also potentially be used to identify outliers in observational data sets, and is a promising method for identifying unique or interesting aerosol populations automatically in large data-sets from research field campaigns. In addition, we were able to achieve high separability between the aerosols classes when using the latent space representations of the L-II signals as input to a supervised classification algorithm. In particular, the distinct separation observed between the two classes of iron oxide aerosols was promising, suggesting that differences between the response of these two iron oxide aerosols in the SP2 could be further exploited to improve the detectability of these aerosols.

When comparing the linear and non-linear unsupervised machine learning approaches (PCA vs. the 2D and 3D VAEs), we found that the 2D and 3D VAEs did perform slightly better in terms of finding lower dimensional representations of the L-II signals that demonstrated higher within class similarity (Section 5) and improved downstream classification (Section 6). However, the advantage was relatively small, indicating that more interpretable PCA analysis could already provide many useful insights when analyzing L-II signals. More sophisticated unsupervised and semi-supervised learning approaches (as we comment on in the last paragraph of this section) could provide additional advantages over the VAE and PCA approaches, however, and warrant future research.

Black carbon and other aerosols that are detected by the SP2, are operationally defined– that is, we classify aerosol by composition based on their response in the instrument and their similarity to laboratory-based proxies for these aerosols.

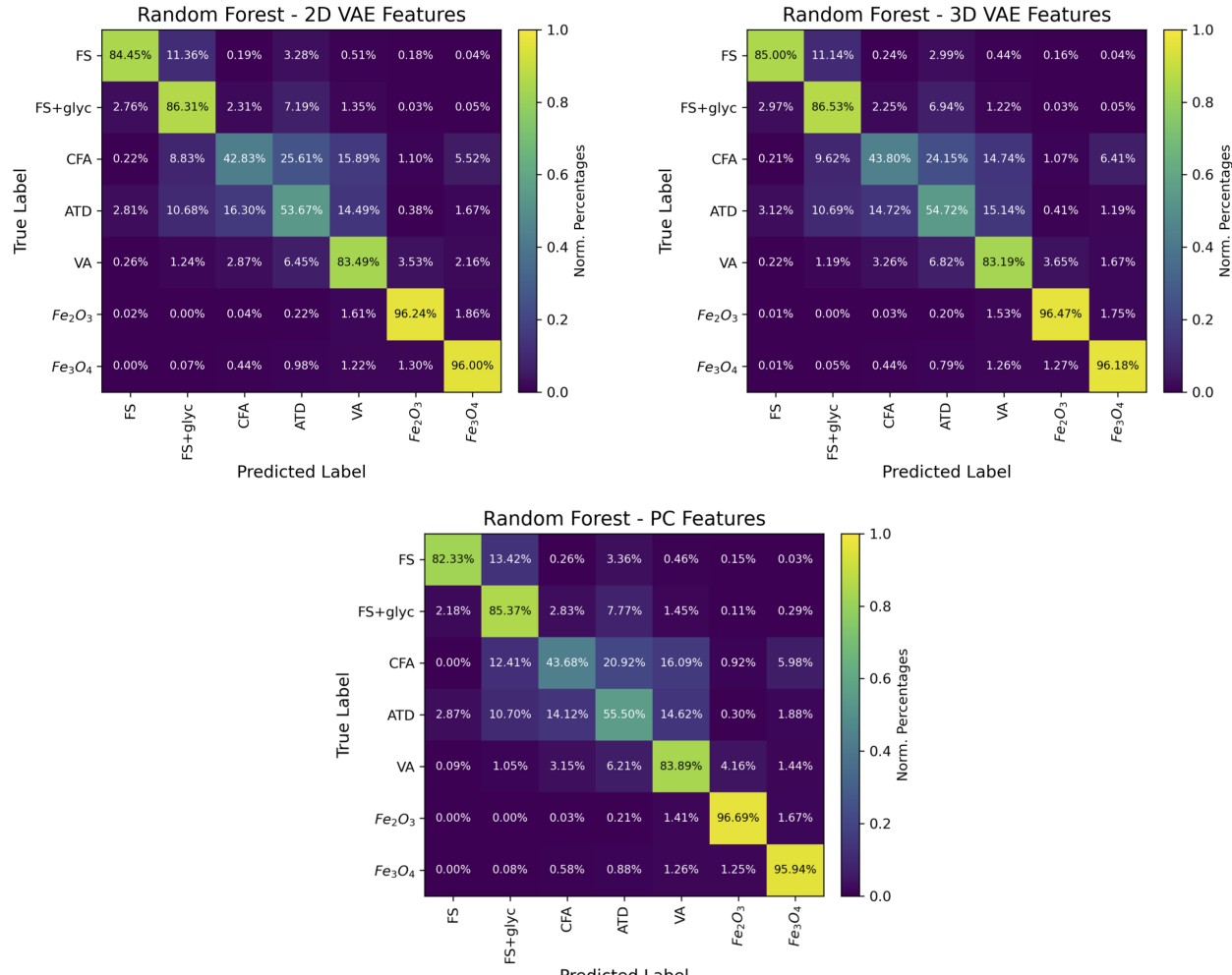

**Figure 11. Performance of RF on classifying different types of absorbing aerosols observed by the SP2 based on their latent representations.** Top left: Confusion matrix for 2D latent space RF. Top right: Confusion matrix for the 3D latent space RF. Bottom: Confusion matrix for the PCA analysis

445 However, populations of aerosols in the atmosphere, even those from the same emission sources, will have a distribution of characteristics in terms of their composition, optical properties, and sizes. These differences contribute to variations in the response of these aerosols in the SP2. The unsupervised machine learning analysis that we discuss in this paper demonstrates one method that can be used to quantitatively assess variations between L-II signals, by comparing their similarity in a lower dimensional representation space.

450  In addition to visualizing and quantifying the variability of the populations of aerosols detected by the SP2, the dimensionality reduction approaches we have explored here could potentially be used as a method to identify outliers in populations of

aerosols. One challenge with using the SP2 to identify $FeO_x$ aerosols is that some dust-like aerosols have similar $I_{max}$ vs. CR values as $FeO_x$ (Lamb, 2019; Lamb et al., 2021). This makes it challenging to quantify $FeO_x$ mass loadings in atmospheric conditions, particularly in remote regions where particles need to be differentiated on an individual, rather than population, basis (Lamb et al., 2021). Outlier detection using lower dimensional representations of the L-II signals could be used as an alternative approach to identify dust-like aerosols that may be mis-categorized as $FeO_x$. Due to the random, scattered nature of the L-II signals for dust-like particles, reducing the occurrence of false positives when identifying ambient aerosols as $FeO_x$ would improve the accuracy of $FeO_x$ mass loading measurements.

Furthermore, outlier detection could also potentially be used as an approach to identify interesting populations of aerosols in ground-based or airborne field observational datasets in an unsupervised manner. A number of recent studies have used observations from the SP2 to identify unique populations of aerosols from their L-II signals, including tar brown carbon (Corbin and Gysel-Beer, 2019), iron oxide aerosols (Lamb et al., 2021), or black carbon associated with pyrocumulonimbus (Katich et al., 2023). By automatizing the detection of interesting or unique populations of aerosols that the SP2 detects, these methods can help to identify or characterize aerosols from different atmospheric sources that may not be evident with traditional SP2 L-II analysis approaches. However, because outliers may correspond to L-II signals that fall far outside the distribution the VAE was trained on, their latent embeddings can become unreliable. This highlights an important open question regarding the robustness of latent-space methods for outlier detection for the SP2, which warrants future research. The lower dimensional representations may also provide additional information about the shape of the L-II signals (beyond coating state or refractory aerosol mass) that can be linked to the physio-chemical characteristics of the aerosol particles, as we demonstrated in Figure 5.

There are a number of promising future research directions in terms of applying more advanced unsupervised and semi-supervised machine learning methods to the L-II signals from the SP2 than we have explored here. In particular, contrastive learning is a promising approach for identifying latent space representations that most meaningfully separate aerosols of different classes (Severson et al., 2019; Abid and Zou, 2019). This approach could improve classification when training on labeled laboratory data sets and applying to atmospheric data sets. Similarity metrics, such as the cosine similarity metric that we have discussed here, can be used to determine how similar the SP2 response of aerosols are, enabling more quantitative comparison across field campaign data sets and regions of the atmosphere (Levy et al., 2024). Future research should also further explore how instrument configuration will impact learned representations from the SP2 signal, such that observational data sets across field campaigns and from different instruments could be meaningfully combined towards improved SP2 analysis.

*Code and data availability.* Jupyter notebooks to reproduce all of the figures and analysis described in this paper are available at https://github.com/kdlamb/SP2-VAE. Labeled, machine learning ready data sets for the SP2 L-II signals saved as .npy arrays are available at https://doi.org/10.5281/zenodo.15800436.

*Author contributions.* Conceptualization: K.D.L. Methodology: A.D; K.D.L. Data curation: K.D.L. Data visualization: A.D.; K.D.L. Writing original draft: A.D; K.D.L. All authors approved the final submitted draft.

*Competing interests.* The authors declare they have no competing interests.

485 *Acknowledgements.* K.D.L. acknowledges support from the Zegar Family Foundation and the NSF LEAP Center at Columbia University. We thank the two anonymous reviewers, whose comments and suggestions have significantly strengthened our manuscript.

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
