# Peer review of "Unsupervised Classification of Absorbing Aerosols Detected by the Single Particle Soot Photometer"

_EGUsphere, 2025_

## Author Comment (AC1)

We thank both reviewers for their careful reading of the manuscript and their insightful comments, which have encouraged us to make revisions that have significantly strengthened the analysis. We provide a detailed response to their comments below, with the original comments from the reviewers in **black**, our responses in **blue**, and changes to the manuscript in **green**. Line numbers correspond to the revised manuscript. Based on the reviewers' comments, we have made the following major changes to our analysis and the manuscript:

1) We updated the discussion and analysis of the latent space representations to provide more connection between the learned latent variables and typical signal analysis methods.
2) We have added a comparison between the VAE analysis and a PCA analysis to demonstrate the potential value added by the non-linear dimensionality reduction method
3) We have replaced the section on outlier detection with a section on using cosine similarity as a metric to compare the within-class and between-class similarity of L-II signals.

We have also updated the github repository so that our analysis can now be easily reproduced using Google Colab.

\*\*\*\*\*\*\*\*\*\*\*\*\*\*\*\*\*\*\*\*\*\*\*\*\*\*\*\*\*\*\*\*\*\*\*\*\*\*\*\*\*\*\*\*\*\*\*\*\*\*\*\*\*\*\*\*\*\*\*\*\*\*\*\*\*\*\*\*\*\*\*\*\*\*\*\*\*\*\*\*\*\*\*\*\*\*\*\*\*\*\*\*\*\*\*\*\*\*\*\*\*\*\*\*\*\*\*\*\*\*\*\*\*\*\*\*

**Reviewer 2:**

This study presents a new method for classifying aerosol particles based on the Laser-Induced Incandescence (L-II) signals using unsupervised machine learning. The author applied Variational Autoencoder (VAE) to analyze L-II signals and compress into a lower-dimensional latent space. This approach is an improvement because it removes the need for manual feature engineering. The paper is well-structured and easy to follow. The introduction provides sufficient background on the SP2 and the limitations of previous classifying methods. The methods section clearly describes the dataset, data preprocessing, and the VAE model. Despite these strengths, there are several issues that must be addressed before publication.

We thank the reviewer for their thoughtful comments, and their positive feedback.

**General comments:**

1. **Physical Interpretation of Latent Space:** While the VAE approach is a powerful tool for classifying aerosol types, the discussion on the physical interpretation of the latent space (e.g., $z_1 - z_4$) feels underdeveloped. The connection between the latent representation and blackbody temperature (Figure 3) is fascinating and a key finding. What specific features of the L-II signals (e.g., peak sharpness, symmetry, or decay rate) are being captured by these latent variables? A more thorough discussion linking the distributions in Figures 5

and 6 to the microphysical properties of the different aerosol types (BC, FeOx, dust) would significantly strengthen the paper's scientific contribution.

We have now included additional analysis of the decoded signals from the latent space manifolds, which demonstrate what specific types of variability are being picked up by the variational autoencoder in Figure 7, shown below.

[Figure]

**Figure 7: Signal Analysis of Latent Manifolds for the L-II Signals**. The main peak location, peak leading edge location (defined as the leading edge of the largest peak at half maximum), and full-width half maximum for the decoded L-II signals for Ch. 0 (top row) and Ch. 1 (bottom row). The peak location, leading edge location, and FWHM width are all given in terms of digitalized time points (out of 400 for the NOAA SP2).

We have added the following discussion to the manuscript in lines 293 - 301:

To more quantitatively assess the types of variability learned by the VAEs for Ch. 0 and Ch. 1, we analyze the decoded signals associated with each decile, using the scipy package (Virtanen et al. 202). We use the find_peaks function in scipy to find the largest peak in each time series, and then determine the peak location (in terms of digitized time points). We also use the peak_widths function to determine the width of the full width half maximum (FWHM) of this peak, as well as

the digitized time point of the leading edge of the peak at half maximum (which we refer to as leading edge location). The values associated with the decoded signals of the latent manifolds for both Ch. 0 and Ch. 1 are shown in Figure 7. For Ch. 0 in particular it is clear that both the leading edge location and the FWHM are strongly correlated with $z_2$. For Ch. 1, the peak location and leading edge location have their highest values (associated with later times in the L-II signals) towards the center of the manifold. The signals at the center of the manifold are also associated with more narrow peaks in terms of their FWHM.

2. **Outlier Detection and Ambient Data:** The claim that outlier detection can be useful for characterizing aerosols from various sources (Lines 293-301) seems a strong assertion. While this is a promising application, the current study, which uses laboratory-generated data, does not provide sufficient evidence to support this claim for ambient atmospheric observations. To make this point more convincing, the authors would need to analyze real atmospheric data. I recommend toning down this claim to a more cautious statement about the potential for this method to be applied to ambient data in future work.

We have now replaced the section on outlier detection to include a calculation of the cosine similarity between the different embedding the signal (2D VAE, 3D VAE, and PCA) in order to demonstrate which classes show the most similarity to one another in Figure 8. We have updated the discussion to suggest that outlier detection will be a promising future direction of research in using unsupervised machine learning for analysis of the L-II signals from the SP2.

[Figure]

[Figure]

[Figure]

3. **Figure Readability:** Overall, the font size for text within the figures, including labels and legends, is too small and difficult to read. The marker sizes in the legends are also unclear, making it hard to distinguish between different aerosol types. I would recommend that the authors increase the font size and marker size to improve the readability of all figures.

We have now updated the font size throughout the manuscript so that figures are more readable.

**Specific comments:**

1. Lines 29-30: Please re-check the citation for Moteki and Kondo (2010). This paper does not focus on a field study of rBC. The citation should be removed or replaced with a more relevant reference.

Thank you for pointing this out. We have replaced the citation with Moteki et al. 2014 instead.

Moteki, N., Y. Kondo, and K. Adachi (2014), Identification by single-particle soot photometer of black carbon particles attached to other particles: Laboratory experiments and ground observations in Tokyo, *J. Geophys. Res. Atmos.*, 119, 1031–1043, doi:10.1002/2013JD020655.

2. Line 78: The chemical formula of Iron (IV) should be corrected to Iron (II, III).

Thank you. Updated.

3. Line 131: The text should be corrected from Ch. 0 to Ch. 1.

Updated.

4. Figure 2: Please correct "Fe203" and "Fe3O4" to "$Fe_2O_3$" and "$Fe_3O_4$". Additionally, please correct "Schwartz et. al 2006."

We have updated the figure to fix the typos.

5. Figure 2: In the center upper panels, "Class 1" and "Class 2" are not defined. Could you clarify what these classes represent?

We have now added the class labels in Table 1.

6. Line 216: Please correct the spelling of "Fe*Ox*". I would recommend the thorough check of the entire manuscript.

We have checked the spelling throughout the manuscript.

7. Figure 4: Why are "z1" and "z3" denoted in x labels and "z2" and "z4" denoted in y labels? It seems that they should be "time" and "signal amplitude."

Here we are denoting the direction of variability sampled for the latent manifold. By sampling along the deciles of the distributions of z1 and z2 variables (in the left panels) and the deciles of the distribution of z3 and z4 (in the right panel), we visualize characteristic L-II signals at each of those points (using the trained decoder to map back to the high dimensional time series space). We have now added the values of the latent variables that are decoded in each subplot along the x and y-axis so that this is clear.

To clarify this, we have updated the caption in (now) Figure 6 to read:

**Latent Manifolds for the L-II Signals.** By sampling along the deciles of a gaussian distribution for the 2 latent variables from each channel, we visualize characteristic L-II signals using the trained decoder. These learned latent manifolds provide an overview of the modes of variability described by the latent representations of the two channels. **Left: Latent Manifold for Channel 0, showing the variability represented by z1 (x-axis) and z2 (y-axis). Right: Latent Manifold for Channel 1, showing the variability represented by z3 (x-axis) and z4 (y-axis).**

8. Figure 4: The "Noise" signals are not explicitly defined. It would be helpful to provide a clear definition of "Noise" signal in this context.

Because the signals are normalized between 0 and 1, this part of the latent space for Ch. 0 appears to pick up relatively small scattering signals, where the noise on the baseline is comparable to the height of the peaks in the signal. We have added this point to the text in Lines 289-292:

Because the signals are normalized between 0 and 1, the top left part of the latent space for Ch. 0 appears to pick up relatively small scattering signals, where the noise on the baseline is comparable to the height of the peaks in the signal.

9. Lines 242–244 and Figure 5: The authors state that there is "significant overlap" between FS and FS+glyc. However, the distributions for Ch 1 (z4 vs z3) appear to be different.

This is a great point, and it is potentially due to differences in the thermal lensing between the coated and uncoated FS particles. While a significant portion of the latent space shows an overlap between the particles, there is also a population of FS+glyc signals that show up with high density in the top right part of the Ch. 1 latent space. We have added this point to the text in lines 306-310:

However, there are also regions of high density for the Ch. 1 embeddings of the FS+glyc samples that are not occupied by the FS samples. Similarly, for the two iron oxides, $Fe_2O_3$ has a higher density of points near the origin and for positive values of $z_4$ when compared to $Fe_3O_4$. This suggests that while there are some similarities between the latent embeddings for the two types of FS and for the two iron oxide aerosols, there are some notable differences in terms of their response in the SP2. We quantify this similarity further in Section 5.

10. Line 278: It seems that "outlier increases" should be corrected to "outlier decreases."

Good point. Based on the major comments from both reviewers, we have chosen to focus on quantifying within-class and between-class similarity, and have removed the section on outlier detection.

11. Figure 7: The scatter plots of z4 and z3 in the left two panels appear to be almost identical. It would be more efficient to include only one panel. Additionally, please clarify what the dashed lines indicate and confirm that they correspond to the selected outlier markers.
12. Figure 7: The signals of Channel 3 are difficult to interpret for unfamiliar reader to interpret. It would be helpful that author to include an example of a normal signal of Channel 3 as well as Channels 0 and 1 showed in Figure 2.

We have now removed Figure 7 and replaced this outlier analysis with a comparison of the cosine similarity of the embeddings of the different aerosol classes.

13. Lines 281–285: Please provide a physical explanation for why these specific outliers occurred. For example, was the $Fe_3O_4$ outlier due to the multiple detection of particles?

For the $Fe_3O_4$ particle (top panel, right), this particle was likely considered an outlier because two particles passed through the beam during the same triggering window, which is evident in the inset plot for Ch. 0. For the CFA particle (bottom panel, right), this L-II signal was identified as an outlier because the particle was very large (clear from the saturated peak in Ch. 0 and the irregular signal in Ch. 4) and incandescent occurred relatively late during the time the particle was passing through ND:YAG laser beam (clear from the incandescence peaks in Ch. 1 and 2 being shifted to the far right in the time series signals).

As mentioned in the previous comments, we have chosen to remove the section on outlier detection and instead include a section on quantifying the variability in aerosol populations using their L-II embeddings.

---

## Author Comment (AC2)

We thank both reviewers for their careful reading of the manuscript and their insightful comments, which have encouraged us to make revisions that have significantly strengthened the analysis. We provide a detailed response to their comments below, with the original comments from the reviewers in **black**, our responses in **blue**, and changes to the manuscript in **green**. Line numbers correspond to the revised manuscript. Based on the reviewers' comments, we have made the following major changes to our analysis and the manuscript:

1) We updated the discussion and analysis of the latent space representations to provide more connection between the learned latent variables and typical signal analysis methods.
2) We have added a comparison between the VAE analysis and a PCA analysis to demonstrate the potential value added by the non-linear dimensionality reduction method
3) We have replaced the section on outlier detection with a section on using cosine similarity as a metric to compare the within-class and between-class similarity of L-II signals.

We have also updated the github repository so that our analysis can now be easily reproduced using Google Colab.
* * *
**Reviewer 1:**

The paper by Doshi and Lamb introduces an unsupervised machine learning approach to better understand the structure of absorbing aerosols using L-II signals from the SP2. Using a variational autoencoder (VAE) the authors are able to extract a compressed latent feature vector of the L-II signals, and use this for outlier detection and enhanced identification of distinct aerosol populations (even outperforming previous tests using significant feature engineering). The paper is generally well written and I appreciate the conciseness of everything. Before fully recommending the paper for publication, I have a handful of questions/comments I'd like to see addressed surrounding latent feature physical interpretations, the dimensionality reduction methodology, outlier detection approach, and generalizability. Further, several of the figures should be updated to match the specifications set forth by EGU (i.e., enhanced text/label size throughout and improved color choices for visibility) to improve general readability.

We thank the reviewer for their careful reading of the manuscript and for providing very useful critical feedback that has helped us to improve our analysis and the robustness of our results.

**General Comments:**

1. **Physical Interpretations:** At the core of this research project is the latent feature vector [Z1, Z2] produced by the VAE, however it wasn't clear to me throughout the analysis what these values represent physically? I.e., what does this compressed representation mean wrt. aerosols? This is always a challenge in DR projects, but I think it is important to consider since this vector is used throughout the remainder of the results for visualization/interpreting different classes/outlier etc., and we would like to have some confidence that it is learning a real physical feature in the L-II signals, and not some

spurious measurement artifact from the SP2 for instance. I understand Section 3 talks a bit towards this point, but I'd like to see more detail from the authors examining this in more detail, and if possible, provide some validation of their interpretation to ancillary data (if available).

We have now expanded Section 3 to further discuss the physical interpretability of the latent space dimension. In particular, we added an additional figure (Figure 7) to the manuscript analysing the latent manifold signals to indicate what types of variance are encoded in the latent space variables.

[Figure]

**Figure 7: Signal Analysis of Latent Manifolds for the L-II Signals**. The main peak location, peak leading edge location (defined as the leading edge of the largest peak at half maximum), and full-width half maximum for the decoded L-II signals for Ch. 0 (top row) and Ch. 1 (bottom row). The peak location, leading edge location, and FWHM width are all given in terms of digitalized time points (out of 400 for the NOAA SP2).

We have added the following discussion to the manuscript in lines 293 - 301:

To more quantitatively assess the types of variability learned by the VAEs for Ch. 0 and Ch. 1, we analyze the decoded signals associated with each decile, using the scipy package (Virtanen et al. 202). We use the find_peaks function in scipy to find the largest peak in each time series, and

then determine the peak location (in terms of digitized time points). We also use the peak_widths function to determine the width of the full width half maximum (FWHM) of this peak, as well as the digitized time point of the leading edge of the peak at half maximum (which we refer to as leading edge location). The values associated with the decoded signals of the latent manifolds for both Ch. 0 and Ch. 1 are shown in Figure 7. For Ch. 0 in particular it is clear that both the leading edge location and the FWHM are strongly correlated with $z_2$. For Ch. 1, the peak location and leading edge location have their highest values (associated with later times in the L-II signals) towards the center of the manifold. The signals at the center of the manifold are also associated with more narrow peaks in terms of their FWHM.

2. **Dimensionality Reduction Methods:** Building on my previous point, VAEs are great at capturing nonlinear relationships and for providing a smooth manifold, but the latent embeddings are typically less interpretable that the EOFs presented by a PCA for instance. While I understand this problem likely benefits from nonlinear DR like VAEs, UMAP etc., did you attempt to fit a PCA to the same data? What EOFs were produced and how did they differ from those using the VAE? This should be easy to fit, and I feel would be a useful/interesting comparison for this project to include as it provides a linear baseline for comparison, motivating the need for a more computationally expensive technique like the VAE.

We have now included the discussion of the PCA analysis of the SP2 signals, and included a Jupyter notebook of this analysis in our github repository. The characteristic EOFs for the first 5 principal components for Ch.0 and Ch.1 are shown below (Figure 3):

[Figure]

**Figure 3. First 5 Principal Components of the L-II signals.** Top row: Ch. 0 (Scattering channel). Bottom row: Ch. 1 (Incandescent channel)

We have expanded the discussion throughout the paper to include the comparison between the VAE and a PCA analysis. In particular, we have now added additional figures of the aerosols in terms of the first two PCs of Ch. 0 and Ch. 1 (Figure 4). We also compare PCA analysis with the

2D and 3D VAE in Figures 10 (the cosine similarity metric between classes) and Figures 11 (the random forest prediction of aerosol class).

3. **Outlier Detection:** I like this idea for outlier detection (Section 4) using the latent manifolds from the VAE. However, I am not totally convinced that the centroid + euclidean distance approach is the most robust approach for this. As the authors illustrate in Figs. 5/6, the manifolds from the latent vector don't produce circular/spherical distributions across the embedding plane. When dealing with continuous real-world observations, we instead typically see these much more abstract and diverse shapes, where a simple euclidean distance from the centroid might not capture a true outlier, but instead, another relevant mode in the data. I would recommend the authors consider including a discussion/comparison to other approaches for outlier detection (e.g., HDBSCAN) which looks at the density of the cases and how they are distributed across the scene.

We explored using HBSCAN, and found it promising but not conclusive in terms of unsupervised identification of various aerosol classes.

When applying HDBSCAN to the 2D latent features from both the scattering and incandescent channels as well as the color temperature ratio, the incandescent maximum, and the scattering maximum, with a minimum cluster size of 50, we did find promise in unsupervised identification of iron oxide aerosols from fullerene soot. However, we did not find total separability in terms of identifying the individual classes, as shown in Figure R1. However, HDBSCAN did do a good job of identifying aerosols that seemed to be outliers in color temperature vs. incandescent maximum space (Figure R2), suggesting that this is a promising approach that should be explored in more detail in future research.

We agree that the Euclidean distance approach has its limitations in terms of identifying outliers from the latent space manifolds, given their complex structures. We have therefore chosen to deemphasize the outlier detection analysis (and instead discuss this as a promising future research direction). We instead replaced this section with a new analysis of the cosine similarity of the embeddings from the various dimensionality reduction techniques (PCA, 2D and 3D VAE) within and between aerosol classes, which provides a quantitative metric for how similar (or different) aerosols within and between each class are in terms of their response in the SP2.

[Figure]

**Figure R1**: Class vs. cluster identification by HDBSCAN.

[Figure]

**Figure R2:** Incandescent maximum vs. color ratio space for the L-II signals, colored by aerosol class (left) and HDBSCAN cluster label (right). Black points in the HDBSCAN cluster labeled plot indicate L-II signals that were identified as outliers (i.e. not belonging to any of the HDBSCAN clusters).

4. **Generalizability:** Since this work is based on laboratory data, do you expect the results of the VAE manifolds to change if applied to actual atmospheric observations? How robust is this to different regional climates/periods, and how stable would you expect the VAE manifold to remain? It would be nice to see some additional details or some text in the discussion addressing the applicability of this approach and these results to new, unseen data.

The VAE will depend on the data that it is trained on, as well as the training process itself, since the initial weights of the neural networks are randomly selected. However, once trained, the VAE can be applied in its inference mode to any new data set. Thus the VAE could be trained on laboratory samples and then could be applied to ambient measurements, allowing a comparison between the latent representations of ambient aerosols and laboratory proxies.

We have now added this paragraph to Section 2.3 in lines 225-231:

Since the specific latent representations are learned from the distribution of data that the VAE is trained on, the VAE in general will not learn the same latent representation from a new data set. Training on a representative sample of laboratory measurements (as we do here) can provide a wide range of possible samples for the generative model. However, once trained, the VAE can be applied in its inference mode to any new data set, providing the same mapping between input signals and lower dimensional representations. For example, the VAE could be trained on laboratory samples and then can be applied to ambient measurements, allowing a comparison between the latent representations of ambient aerosols and laboratory proxies.

**Specific Comments:**

1. **Title:** I would remove "(VAE)" from the title

We have updated the title to "Unsupervised Classification of Absorbing Aerosols Detected by the Single Particle Soot Photometer", as we have now expanded the analysis to include PCA.

2. **Figure 1:** Can you add some additional details to the figure description explaining the SP2 (outside of just the citation).

We have expanded the description in the caption to read:

Aerosols sampled by the SP2 are introduced into the cavity of an ND:YAG laser (the beam is indicated as red curved lines), where they scatter light and potentially incandesce in the laser beam. Four photomultiplier tubes (indicated in the figure as the red detector, blue detector, and scattering light detectors) acquire time series of the light scattered or emitted by the aerosol particles as they interact with the laser beam.

3. **Line 36:** Should be "incandesce", right?

Yes, corrected.

4. **Lines 48-49:** This felt like a really abrupt change in the introduction, moving from the SP2 to unsupervised machine learning. I would suggest reworking these paragraphs to flow more clearly/logically.

Great point. We have added an additional paragraph to discuss the typical usage of the SP2 in measuring aerosols during field campaigns, and the large amount of data that the SP2 acquires, which requires automated data processing methods in lines 50-56:

The SP2 is typically used in both ground-based and airborne field campaigns to monitor ambient aerosol particles (e.g., Schwarz et al. 2006, Schwarz et al. 2010, Lamb et al. 2018, Lamb et al. 2021, Katich et al. 2023). During a typical research flight in a source region, the SP2 might acquire signals for ~4 million individual aerosol particles, necessitating automated detection techniques to process L-II signals from the instrument. Typical automated methods to process L-II signals use curve-fitting techniques and calibrations from laboratory studies to determine aerosol particle mass and coating thickness assuming a Mie-theory core shell framework (e.g. Gao et al. 2007, Schwarz et al. 2006, Schwarz et al 2010). As an alternative approach to traditional SP2 data processing techniques, Lamb (2019) alternatively explored a supervised machine learning approach.

5. **Lines 49-63:** I recognize this is a follow-on from Lamb (2019), but are there other references you could include throughout this section looking at supervised/unsupervised ML in aerosols to better motivate this work? The paper would likely benefit from broadening the current set of references to better situate itself in current literature.

Yes, this is a good point. We have now included a paragraph in the introduction on the recent use of machine learning for atmospheric observations from other types of in situ aerosol and cloud particle instrumentation besides the SP2 in lines 57-66:

Machine learning is increasingly being applied to other types of specialized instrumentation used to detect in situ aerosol and cloud particles, providing new insights into the properties of atmospheric aerosol and cloud particle populations in large data sets. While initial efforts have focused on supervised machine learning (for example, for the Wideband Integrated Bioaerosol Sensor (WIBS) (Rusket et al. 2017; 2018), the Particle Analysis by Laser Mass Spectrometry (PALMS) instrument (Christopoulus et al. 2018; Zawadowicz et al. 2017), and for cloud particle imager probe images (e.g. Przybylo et al. 2022), more recent efforts have also explored unsupervised and self-supervised machine learning approaches, sometimes in combination with co-sampled environmental observations (e.g. Allwayin et al. 2024; Ko et al. 2025). As large datasets of atmospheric observations from in situ observations are increasingly being compiled across field campaigns and sampling conditions, these methods promise to lead to new physical insights into the atmospheric processes that underlie these observations (Lamb et al. 2025).

Allwayin, N., Larsen, M.L., Glienke, S. and Shaw, R.A., 2024. Locally narrow droplet size distributions are ubiquitous in stratocumulus clouds. *Science*, *384*(6695), pp.528-532.

Ko, J., Govindarajan, H., Lindsten, F., Przybylo, V., Sulia, K., van Lier-Walqui, M. and Lamb, K., 2025. Understanding Ice Crystal Habit Diversity with Self-Supervised Learning. *Tackling Climate Change with AI Workshop, Conference on Neural Information Processing Systems*, 2025. *arXiv preprint arXiv:2509.07688*.

Lamb, K.D. and Singer, C. and Loftus, K. and Morrison, H. and Powell, M. and Ko, J. and Buch, J. and Hu, A. and van Lier Walqui, M. and Gentine, P. Perspectives on Systematic Cloud Microphysics Scheme Development with Machine Learning. *Journal of Advances in Modeling Earth Systems* (under review), 2025.

Przybylo, V.M., Sulia, K.J., Schmitt, C.G. and Lebo, Z.J., 2022. Classification of Cloud Particle Imagery from Aircraft Platforms Using.

6. **Lines 57-59:** You write "unsupervised machine learning" three times in three lines and it reads as quite repetitive, I would also recommend reworking this portion of the introduction to flow more cleanly.

We have reworked this paragraph (lines 75-84) to connect more clearly to the previous paragraph on limitations of supervised methods:

On the other hand, unsupervised machine learning algorithms discern inherent patterns and correlations in data sets, without requiring predefined categories. These methods have not previously been applied to the problem of classifying L-II signals, and they may address the limited generalization performance of supervised methods to ambient aerosol populations where laboratory proxies are not available or only partially replicate the characteristics of ambient aerosols. Here we explore how unsupervised machine learning can be applied to L-II signals from the SP2, with the goal of identifying different populations of aerosols based on the information in their L-II signals alone. We also investigate whether unsupervised methods can provide insights into the variability of aerosols of different types based on their L-II response in the SP2. Our analysis provides insight into the amount of independent information that can be gained from L-II signals in terms of identifying the composition of refractory aerosols that reach detectable incandescence in the SP2, and also provides a starting point for future studies on how data-driven methods might provide new insights into the sources and variability of aerosol populations.

7. **Line 67:** I know you define VAE in the abstract, but I typically recommend redefining all abbreviations within the text itself, so it can stand on its own.

Updated.

8. **Lines 153-155:** I am less familiar with the experimental setup for aerosol-specific model training, but are there any concerns about overfitting from temporal autocorrection using a random 50/25/25 split over a segmented approach?

The individual aerosol detection events (what we are referring to here as the L-II signals) in the SP2 are determined by when the voltage on the scattering detector passes a specific threshold

value due to an aerosol particle passing through the center of the laser beam. Based on this trigger, a predetermined time series length of 80 microseconds (400 time point signal) is saved as the signal associated with an individual aerosol particle. This pre-processing is done before the machine learning algorithms are applied, and thus we expect each L-II signal to represent an independent triggering event. Therefore we do not expect any temporal auto-correlation will play a role in the analysis of these data sets, as each L-II signal represents an individual aerosol detection event, and we treat each signal as an independent sample.

To clarify this in the text, we have added this sentence to the Dataset description in lines 97-99:

Each L-II signal represents an individual aerosol detection event in the SP2, and we treat each signal as an independent sample in applying the machine learning algorithms.

9. **Figure 2:** I like this figure, but the text is really small/hard to read in its current state

We have updated the text size so that it is easier to read.

10. **Line 232:** You mention hyperparameter selection here, but don't go into details about the process. What approach was used/what search space was evaluated? I summary table with final tuned values would also be a useful addition to consider.

We were referring to varying the size of the latent space from 2 - 5 variables, which we have discussed in detail in the manuscript. We expanded the discussion here to now include an estimation of the intrinsic dimension of the time series for Ch. 0 and Ch. 1, which further solidifies our conclusion that 3 variables is likely the maximum number needed for the VAE to capture the variance in our data set in lines 342-346:

We additionally estimated the intrinsic dimension of the latent variables associated with Ch. 0 and Ch. 1 for n=2,3,4, and 5 latent variables. The intrinsic dimension is the minimum number of variables required to represent the variability in the dataset. We used the intrinsic dimension algorithms in sci-kit dimension (Bac et al. 2021), and found that independent of the intrinsic dimension algorithm used, the estimated intrinsic dimension was approximately 3 for the time series associated with both channels.

We did not extensively explore the hyperparameters in the VAE or the random forest because our focus here was on demonstrating the potential value of an unsupervised learning approach for identifying useful features in the SP2 signals that might help to differentiate between different types of aerosols, rather than optimizing performance on this specific data set.

11. **Figures 5/6:** Similar to my previous comment about label size. Also the yellow cases are really hard to see on the white background. It might also be useful to apply a set of coloblind-friendly colors here so the "All classes" figure is more easily digested.

We have adjusted the font size and updated the colors to be more visible.

12. **Figure 7:** Font size comment again

We have updated this section to focus on the cosine similarity calculation of the embeddings in each class, so we have removed this figure from the updated version of the paper.

13. Line 361-362: I appreciate the inclusion of the GitHub link with code for reproducibility, but when examining it, there are no included datasets to test with? Unless I am missing this step, I would recommend taking a subset of your *.npy data depending on data size and making it available on Colab through gdown (or some similar alternative). Then users can simply run your notebook online to reproduce the primary results.

As mentioned in the Code and Data availability section, the data sets used in this study are available in a Zenodo repository as numpy arrays: https://doi.org/10.5281/zenodo.15800436.

We have now refactored the code and set up the github so that the Jupyter notebooks can be opened directly in Colab and the datasets can be immediately downloaded to Colab from Zenodo in order to facilitate reproducibility.

---

## Author Response (AR2)

I appreciate the effort the authors have put forth in addressing reviewer comments from the previous round of revisions. The newly included figures, comparisons with PCA, and cosine similarity analysis has helped reinforce the paper's results, and made for a more compelling read. I also appreciate the effort put forth in improving the figure legibility, as this is now greatly improved. Before I fully recommend the paper for publication, I did have a few minor comments for the authors to consider:

We thank the reviewer for their careful reading of the manuscript and for their additional comments and suggestions. They have further improved the quality and clarity of the manuscript.

General: I'd recommend doing one more read-through for grammar/spacing etc. since a lot of the text has changed. I won't list everything but small errors like a missing space on lines 235-238, also I believe it should be "scikit-learn", should be addressed before publication.

Thanks for the suggestion. We have reread the text and corrected a number of small typos throughout.

Figure 5: The tick labels on the far-right plot colorbar are not properly aligned.

We have updated the tick labels to be properly aligned.

[Figure]

Figure 7: I like the addition of this figure, but I think it is lacking a bit of context. For instance, x/y-axis labels. Further, laid out this way, it feels like we should be comparing between the scattering and incandescent channels, but they colorbars have different scales which makes this challenging.

We have added x and y labels to the axes. We also flipped the orientation from horizontal to vertical to make it clearer that the comparison should be against the latent manifolds shown in Figure 6, rather than between the two channels. We have updated the caption of Figure 7 to reference Figure 6 in order to clarify this.

[Figure]

Also, what is the bright/dark feature at 1.64/0.39 in Ch. 0 and how should we interpret it? It really draws the eye here.

As can be seen in the latent manifold for Ch. 0, the reconstructed Ch. 0 signal corresponding to the latent values of (1.64, 0.39) is due to the second large peak in the signal, which is to the far right in the L-II time-series. Because the initial peak (which triggers the SP2 detection) is very small for this particular signal, the later peak is the one picked out by the peak finding algorithm from Scipy's signal analysis package as the predominant peak. Other reconstructed time series also show evidence of a second peak but they are not the dominant peaks in the time series, and thus are ignored in this analysis. This demonstrates how the VAE is able to pick up some of the more subtle shifts in the L-II signals, when compared with a more naïve application of a peak-finding algorithm that is traditionally used in analyzing the L-II signals from the SP2.

Section 5/Figure 10: This is an interesting analysis, especially the comparison with the PCA. I am curious though, due to the very high pattern correlation across both the 2D/3D VAE matrices, and the PCA matrix (which look basically the same with slightly different magnitudes), what is gained from a nonlinear approach over the PCA? Or is the PCA sufficient here? It wasn't clear what the takeaway was now that you have shifted from a solely nonlinear approach to PCA+VAE.

We added the following paragraph to the Conclusions in lines 436-442:

When comparing the linear and non-linear unsupervised machine learning approaches (PCA vs. the 2D and 3D VAEs), we found that the 2D and 3D VAEs did perform slightly better in terms of finding lower dimensional representations of the L-II signals that demonstrated higher within class similarity (Section 5) and improved downstream classification (Section 6). However, the advantage was relatively small, indicating that more interpretable PCA analysis could already provide many useful insights when analyzing L-II signals. More sophisticated unsupervised and semi-supervised learning approaches (as we comment on in the last paragraph of this section) could provide additional advantages over the VAE and PCA approaches, however, and warrant future research.

Section 2.3: On the generalizability topic, I was more so considering the fact that after training on the lab data, if that model is then used operationally and encounters aerosols outside the convex hull of training data, that the compressed latent vector may no longer contain relevant information, as the model has never seen cases such as these. We've run into issues like this before where we did our best to get the most robust set of training samples possible, but in real world situations, there is a chance to encounter something outside of the training set which the model does not necessarily handle properly. I was hoping to see some discussion around this and whether or not this would be an issue with the aerosol data.

This is the point that we are trying to make in terms of the utility of these approaches for outlier detection (e.g. finding populations that are not well-represented in laboratory data). We have added a caveat to the discussion to acknowledge this point in Lines 465 – 467:

However, because outliers may correspond to L-II signals that fall far outside the distribution the VAE was trained on, their latent embeddings can become unreliable. This highlights an important open question regarding the robustness of latent-space methods for outlier detection for the SP2, which warrants future research.